# Accurate haplotype construction and detection of selection signatures enabled by high quality pig genome sequences

Xinkai Tong[1,2], Dong Chen[1], Jianchao Hu[1], Shiyao Lin[1], Ziqi Ling[1], Huashui Ai [1], Zhiyan Zhang [1] ✉ & Lusheng Huang [1] ✉

High-quality whole-genome resequencing in large-scale pig populations with pedigree structure and multiple breeds would enable accurate construction of haplotype and robust selection-signature detection. Here, we sequence 740 pigs, combine with 149 of our previously published resequencing data, retrieve 207 resequencing datasets, and form a panel of worldwide distributed wild boars, aboriginal and highly selected pigs with pedigree structures, amounting to 1096 genomes from 43 breeds. Combining with their haplotype-informative reads and pedigree structure, we accurately construct a panel of 1874 haploid genomes with 41,964,356 genetic variants. We further demonstrate its valuable applications in GWAS by identifying five novel loci for intramuscular fat content, and in genomic selection by increasing the accuracy of estimated breeding value by 36.7%. In evolutionary selection, we detect *MUC13* gene under a long-term balancing selection, as well as *NPR3* gene under positive selection for pig stature. Our study provides abundant genomic variations for robust selection-signature detection and accurate haplotypes for deciphering complex traits in pigs.

The pig can serve as an excellent human biomedical model and promising xenotransplantation donor due to its similarity to humans in anatomy, physiology, and genetics[1]. More importantly, it delivered more than 100 million metric tons of meat to the world in 2021 (https://www.statista.com/statistics/273232/net-pork-production-worldwide-by-country/#statisticContainer), playing a crucial role in agricultural meat production for human beings. Investigating the genetic characteristics of pigs and the evolutionary mechanism of their important traits will help us make the most of its value in both biomedical model and agricultural aspects. There are numerous genome-wide association studies (GWAS) on growth, meat quality, and disease resistance by low or medium density SNP chips in large-scale pig cohorts[2–6]. Although the cost of SNP chips is less than resequencing technology, their sparse density restricts the application to fine mapping and identification of causal variants for the related traits. In this case, imputing genotypes of SNP chips into genotypes of high-density variant panels by utilizing the haplotype reference panel for the deep genetic dissection of the studied pig phenotypes has become an effective and efficient approach[7]. In this sense, the haplotype reference panel's variant density, phasing accuracy, and diversity are critical for imputation. Several studies have provided pig haplotype reference panels, but they have certain limitations in sample and breed size, variant number, and phasing accuracy[8–11]. Therefore, it is essential to construct a high-quality pig haplotype reference panel with multiple breeds and plentiful individuals for genotype imputation and further genetic investigations.

As a long-domesticated species, the pig has undergone a series of selections for body size, disease resistance, meat quality, etc[12–14]. There are possibly over 730 pig breeds worldwide, of which two-thirds are distributed in China and Europe[15]. These breeds vary widely in stature,

[1]National Key Laboratory for Swine genetic improvement and production technology, Ministry of Science and Technology of China, Jiangxi Agricultural University, NanChang, Jiangxi Province, PR China. [2]College of Life Sciences, Jiangxi Normal University, NanChang, Jiangxi Province, PR China. ✉ e-mail: bioducklily@hotmail.com; lushenghuang@hotmail.com

disease resistance, coat color, behavior, fertility, and other characteristics, which are primarily governed by genetic factors. For instance, *PLAG1* and *BMP2* have been identified as strong candidate genes associated with body size[16–18]. *MUC13* and *RNF216* were associated with disease resistance[4,19]. In domestic pigs, *PLAG1* has been subjected to long-term positive selection[12], and the favorable allele conferring larger stature has a higher frequency in domestic pigs than in wild boars[20]. In contrast, several loci in humans that should theoretically be subject to long-term positive selection did not appear to have a high frequency of advantageous alleles, possibly a consequence of adverse selection on other traits[21–24]. Simultaneously, the polymorphism around these loci is significantly higher than the genome in the remaining regions. In this sense, they can be defined as long-term balancing selection loci[25]. There are common genes under balancing selection that have been identified in mammals previously, such as *HLA*, *ABO*, and *HBB*[21–24,26]. Although several genes under balancing selection have been discovered in studies of pigs[27,28], they were found in breeding populations selected by favorable traits in the short term. For example, balancing selection was identified for a 200 kb deletion affecting the *BMPER* and *BBS9* genes in a breeding population of pigs[27]. However, whether genes under long-term balancing selection specific for pigs exist or not remains elusive. Particularly when utilizing genes under balancing selection in the breeding program, extra considerations should be taken into account to minimize negative impacts.

In this work, we sequence 740 pigs and combine 149 of our previously published pig resequencing data, covering most of the common breeds ($N = 30$) in Asia and parts of the worldwide breeds[26,29–32]. Combining the other high-quality pig genome sequences publicly available, we accurately create a pig haplotype reference panel by taking advantage of haplotype-informative reads, pedigree structure (3 resource populations), and population Linkage Disequilibrium information, by totally using resequencing data of 43 breeds ($N \geq 3$) and 1,096 pigs worldwide with an average sequencing depth of $17.1\times$[33–35]. The established haplotype panel contribute to an increased power of GWAS and the accuracy of genomic selection (GS, predicting the breeding values of offspring). By reanalyzing previously published SNP array data, imputation GWAS (iGWAS) detects five novel loci associated with intramuscular fat (IMF) content. Subsequently, the accuracy of the estimated breeding value (EBV) of IMF is significantly improved ($R^2$ from 0.49 to 0.67). Furthermore, we perform positive- and balancing-selection analyses. Positive-selection analyses identify a candidate gene controlling stature variation. Balancing-selective analyses reveal another balancing selection gene associated with diarrhea- and growth-related traits. Together, the study provides rich genetic variants and high-quality haplotype resources of the pig genome, and we demonstrate their utilization by discovering novel candidate genes for IMF, body stature, and balancing selection.

## Results

### Description of resequencing data and genome variants discovery

We have sequenced whole genomes of 950 samples (889 non-redundant individuals: 740 sequenced in this study and 149 published in our previous studies[26,29–32]) and retrieved 207 resequencing data from the NCBI SRA database, representing in total 43 pig breeds ($N \geq 3$) with an average sequencing depth of 17.1 × (Supplementary Data 1, Supplementary Fig. 1). These breeds are mainly distributed in China and Europe (Supplementary Fig. 2). All individuals could be geographically divided into seven groups: Asian wild pigs, Southeastern Chinese domestic pigs, Northwestern Chinese domestic pigs, Eurasian crossbred pigs, European crossbred pigs, European domestic pigs, and European wild pigs (Supplementary Fig. 3). We used the GATK official pipeline to detect variants and identified 68,166,098 SNPs and Indels (Insertion and deletion, 1-50bp) after hard filtering. Quality control subsequentially executed by the

followings: 1) We excluded duplicated samples with IBS > 0.95 (Identity by state) between each sample pair, resulting in 1,096 non-redundant individuals. 2) We assigned genotypes of variants whose reads coverage is less than six to missing to avoid error-prone detection of heterozygous variants with low-depth sequencing[36]. 3) We also used routine filtering thresholds of variants missing rate ≥ 0.05 and individuals call rate ≤ 0.85 to obtain high-quality genotypes. Finally, 937 individuals ($N \geq 1$, No.breeds = 41; $N \geq 3$, No.breeds = 33) and 41,964,356 autosomal variants were used for further analyses (Supplementary Data 1). The frequency distribution of variants showed that the number of variants decreases as their minor allele frequency (MAF) increases (Supplementary Fig. 4), which is consistent with the view of "rare is common". To validate the accuracy of variant genotyping, we estimated the concordance rate between sequence genotypes and array genotypes (Porcine SNP Genotyping 50k or 60k BeadChip, Illumina) in 325 individuals. The concordance rate ranged from 96.8% to 99.9%, with an average of 98.6% (Supplementary Fig. 5a). Besides, there are 44 trios and 135 duos in our data. We thus estimated the Mendelian Error rate by summarizing error inherited variants in the 179 parent-offspring trios or duos, resulting in varying Mendelian Error rates from 0.06% to 0.6%, with an average of 0.21% (Supplementary Fig. 5b). Both validations demonstrated that the accuracy of variant genotyping elevated with increased sequencing coverage depth (Supplementary Fig. 5).

### Construction of the haplotype reference panel

The 937 high-quality individuals and 41,964,356 autosomal variants (38,483,119 SNPs; 3,481,237 Indels) were used for constructing a haplotype reference panel. We took the following three steps to extract the linkage and LD information for phasing haplotypes: (1) We first established haplotype blocks based on haplotype-informative reads[33]. (2) We then build haplotype scaffolds by including the pedigree structure (44 trios and 135 duos)[34]. (3) Finally, a high-quality haplotype reference panel with 1874 haplotypes was harvested by integrating linked information in haplotype blocks and scaffolds into LD information from populations[35] (Supplementary Fig. 6). To assess the accuracy of the haplotype reference panel, we selected three common commercial breeds, including ten unrelated individuals (Duroc, $N = 4$; Landrace, $N = 3$; Large white, $N = 3$), as imputation targets and the remaining 1854 haploid genomes as reference panel. We randomly selected 50k, 60k, 80k, 100k, and 300k variants at autosomes and masked the remaining unselected sites to mimic chips for target individuals. We imputed the genotypes of masked sites and estimated the correlation and concordance rate between sequenced genotypes and imputed genotypes. The results demonstrated that their accuracies (Mean $R^2$/concordance rate) were 0.89/99.16%, 0.91/99.24%, 0.92/99.41%, 0.94/99.50%, and 0.96/99.71%, respectively (Fig. 1a). The SNPs have slightly higher accuracy than the Indels (Fig. 1a). Because of the generally high error rate in rare variants imputation[37,38], we also estimated the imputation accuracy in rare variants with minor allele frequency ≤ 0.05, showing Mean $R^2$ of 0.83, 0.84, 0.87, 0.88, and 0.92, respectively. Analogously, we selected three common Chinese indigenous breeds (Erhualian, $N = 4$; Bamaxiang, $N = 3$; Laiwu, $N = 3$) as imputation targets. The results demonstrated that their accuracies (Mean $R^2$/concordance rate) were 0.88/98.80%, 0.90/98.94%, 0.91/99.13%, 0.93/99.25%, and 0.96/99.62%, respectively (Supplementary Fig. 7). The imputation accuracies (Mean $R^2$) in rare variants varied from 0.84 to 0.93. The two results implied that the haplotype reference panel has an excellent performance in both common commercial and Chinese indigenous breeds. Moreover, we estimated the imputation accuracy for each breed. The results showed that LW, WDU, DU, LR, and PT have a high imputation accuracy in 41 breeds included in the reference panel (Supplementary Fig. 8). LWU, SUT, BMX, and EHL have a relatively high accuracy compared with Tibetan pigs (GST, YNT, SCT1, SCT2, TT). The imputation accuracy for wild pigs is relatively

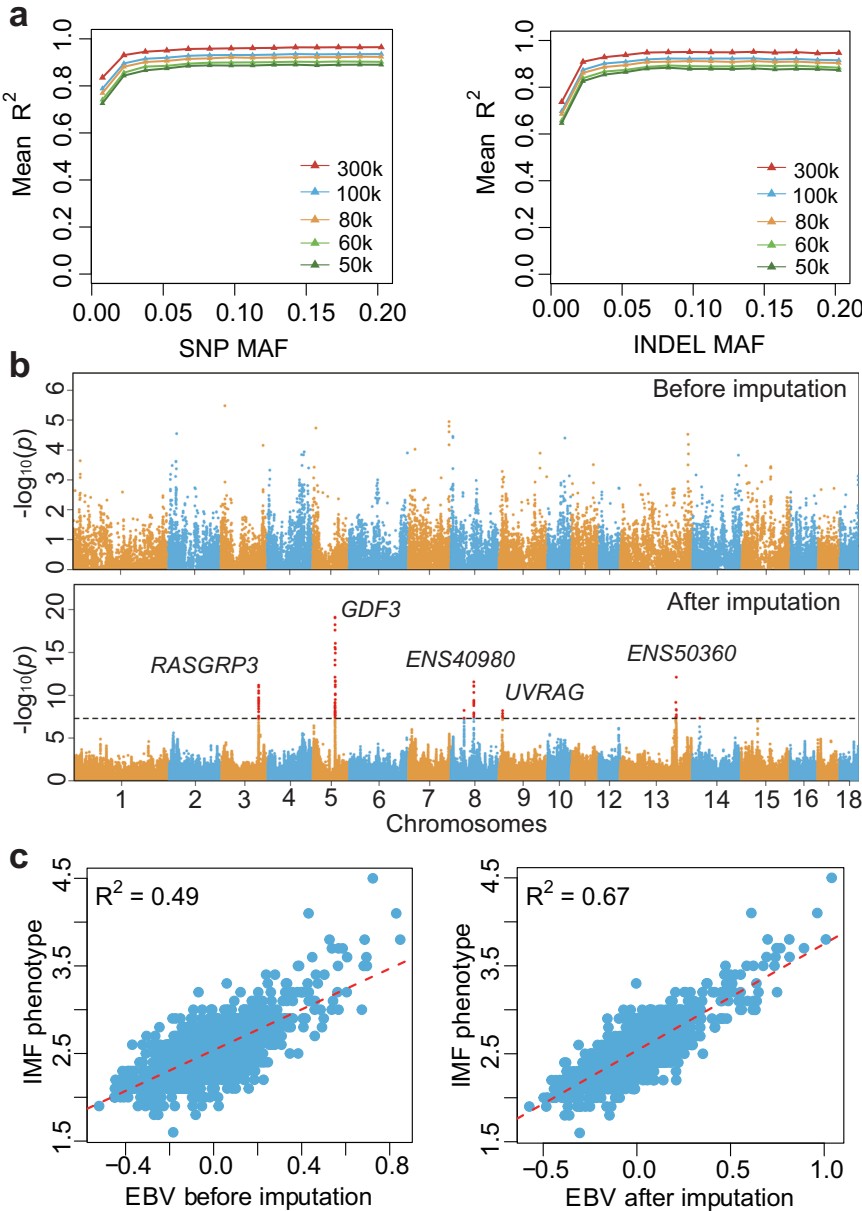

**Fig. 1 | Imputation accuracy using the haplotype reference panel and its application to GWAS and breeding value estimation. a** Imputation accuracy using ten common commercial pigs (Duroc, $N = 4$; Landrace, $N = 3$; Large white, $N = 3$) as targets and the remaining 1854 haplotypes as reference panel. The x-axis shows the minor allele frequency of imputed variants. The y axis shows imputation accuracy measured by average $R^2$ (squared Pearson correlation). The colored triangles represent the $R^2$ under different minor allele frequencies. The 50k, 60k, 80k, 100k, and 300k denote the number of randomly selected polymorphism sites being used for imputation. **b** The manhattan plots of GWASs for Intramuscular fat (IMF) content before and after genotype imputation. The x-axis shows chromosomes. The y-axis shows significant levels of association. Each dot denotes a variant. Wald test is employed to estimate the significance of each variant by gemma v0.98 software. Reported log transformed *P* Values are two-sided and nominal (i.e. not corrected for multiple testing). The genome-wide significance threshold was set to $5 \times 10^{-8}$. ENS40980, ENSSSCG00000040980; ENS50360, ENSSSCG00000050360. **c** The accuracy of estimated breeding value (EBV) before and after genotype imputation. Significant variants at the $P < 0.01$ level from GWAS were used for estimating the breeding value of IMF by BLUP. The x-axis shows the estimated breeding value. The y-axis shows the phenotypic value of IMF. Each dot denotes an individual. Pearson's correlation is employed to evaluate the accuracy of EBV, indicated by an $R^2$ (squared Pearson correlation) between phenotypic value and EBV. The *P* values of correlation coefficient before and after imputation are $5.2 \times 10^{-222}$ and $2.7 \times 10^{-358}$, respectively.

low, such as KRW, VIE, NTLW, SCW, ARSW, NCW, SPAW, ERSW, and ITAW.

In order to rescue the samples being removed for low call rate, we picked up samples with at least an average sequencing depth of 4 × and randomly selected 1,000 k variants to impute missing genotypes. Ninety samples with at least a concordance rate (imputed vs. genotyped) of 97.00% between sequenced genotypes and imputed genotypes were recalled for subsequent analyses (Supplementary Fig. 9).

## Improved power of GWAS by the haplotype reference panel
Previous studies show that the resolution and power for GWAS could be improved by including haplotype reference panel[9,39]. To assess the ability of GWAS power and detection ability by including our haplotype reference panel in an unrelated population with low-density SNP chips. We downloaded both phenotype and genotype (Porcine SNP50 Beadchip, Illumina) of 1,490 Duroc pigs from a published study of IMF[2]. The haplotypes were firstly constructed using 50k array data of

all Duroc pigs. Based on these haplotypes, we imputed missing genotypes using our haplotype reference panel, increasing the number of polymorphic sites from 44,266 (array) to 16,963,108 (imputation) after filtering with minor allele frequency ≥ 0.01. The GWASs were performed using array and imputation genotypes, respectively. Quantile-quantile plots with genome control lambda are shown in Supplementary Fig. 10. We found no evidence of systematic inflation of association test results (lambda = 1.08). The iGWAS identified five novel loci with clear signals which were missed in the chip GWAS (Fig. 1b). Three leading variants are located inside genes *RASGRP3* (3_106129949, $P = 6.7 \times 10^{-12}$), *ENSSSCG00000040980* (8_66234703, $P = 2.8 \times 10^{-12}$), and *UVRAG* (9_10302512, $P = 6.4 \times 10^{-9}$); two leading variants resided 19.9 kb upstream of gene *GDF3* (5_62861562, $P = 7.6 \times 10^{-20}$) and 33.5 kb upstream of *ENSSSCG00000050360* (13_162125842, $P = 7.7 \times 10^{-13}$) (Supplementary Data 2). On the other hand, to strengthen the reliability of these novel loci, we performed haplotype analysis using 50k array data. The haplotypes were constructed using ten SNPs surrounding top SNPs at these novel loci. The analysis of variance (ANOVA) showed that the adjusted phenotype (adjusted by 10 PCAs of genotype) was significantly different (genome-wide threshold, $P < 5 \times 10^{-8}$) among haplotypes at each locus (Supplementary Table 1). Herein, we thought these five novel loci emerged based on phenotypic variation across their haplotypes, which strengthens the reliability of these signals. The above results suggested that our haplotype reference panel has huge application in reanalyzing previously published array data.

## Increased the accuracy of estimated breeding value after imputation

With more novel significant variants and higher power in the iGWAS result by including the haplotype reference panel, we thus try to evaluate the accuracy of estimated breeding values (EBV) using imputed genotypes. Previous studies suggested that genomic prediction would benefit from pre-selecting markers[11,40–43]. We here selected variants at a $P = 0.01$ level from chip GWAS and iGWAS for estimating the breeding value of IMF by BLUP (Best linear unbiased prediction). The correlation ($R^2$) between phenotypic value and imputation EBV ($R^2 = 0.67$) increased by 36.7% compared with chip EBV ($R^2 = 0.49$) (Fig. 1c). Furthermore, we implemented the cross-validated BLUP (cvBLUP) approach to perform a "leave-one-out" analysis for assessing the phenotypic prediction ability of EBV[44]. The result showed that the phenotypic prediction accuracy increased by 20% ($R^2$, from 0.35 to 0.42) using selected imputed whole genome sequencing (WGS) variants (Supplementary Fig. 11).

## Identification of genes under long-term balancing selection

Whole-genome sequencing-based variation discovery in large-scale populations with multiple pig breeds also enables the detection of genes underlying selection. Balancing selection occurs when multiple alleles exist in a population for long periods. These alleles have bidirectional influences on the organism's fitness. This long-term balancing selection is characterized by an excess number of intermediate frequency polymorphisms near the balanced variant[45]. We used the allele frequency correlation method to scan balancing selection loci on autosomes across the following four populations: Southeastern Chinese domestic pigs (SCD, $N = 325$), Northwestern Chinese domestic pigs (NCD, $N = 155$), Crossbred pigs (Cross, $N = 251$), and European domestic pigs (EUD, $N = 210$)[45]. Two significant loci shared by four populations were distinctive (Fig. 2a). One is located at 24.9 Mb on chromosome 7 (SSC7) and overlaps with the major histocompatibility complex (MHC) region. The nearest gene is *SLA-DRB1* (homologous to gene *HLA-DRB1*), which has been identified as experiencing a long-term balancing selection in the vertebrates[23,46]. Another locus with the most potent signal resides at 135.4 Mb on SSC13. To further confirm its balancing selection, we calculated the nucleotide diversity (PI) by a

sliding window of 1 kb in this region. The PI was elevated compared with the surrounding region (Fig. 2b). A positive Tajimas'D also indicate balancing selection[47,48], and we thus estimated the Tajimas'D by a sliding window as same as in PI calculation. As expected, most windows showed a positive Tajimas'D (Fig. 2c). Both results supported that the locus is under balancing selection. The LD analysis of the locus showed that an unambiguous ~8k block exhibited a strong LD with 240 variants distributed in the first two introns of *MUC13* (Fig. 2d, Supplementary Data 3). We then extracted haplotypes located in the block from our constructed haplotype reference panel and found that only two haplotype patterns (Hap1 or Hap2) existed in our analyzed populations (Fig. 2e). The nucleotide sequence divergence between the two 8kb haplotypes was calculated by Jukes-Cantor distance, with an average sequence divergence of 3.04%. Additionally, the frequency of Hap1 stays at an intermediate level (0.58), in line with the characteristics of balancing selection. The frequencies of Hap1 are relatively stable across breeds (Supplementary Table 2).

To trace the emergence time of Hap1 and Hap2, we analyzed the haplotypes of three Asian domestic pigs, three Asian wild pigs, three European domestic pigs, three European wild pigs, one Indonesian wild boar from Sumatra (*Sus scrofa vittatus*), one Visayan warty pig from the Philippines (*Sus cebifrons*), one Javan warty pig from Indonesia (*Sus verrucosus*), one common warthog from Africa (*Phacochoerus africanus*), and three Pygmy hogs from India (*Porcula salvania*). The phylogenetic relationship of haplotypes was assessed by a maximum-likelihood tree. We found that all Hap2 of European and Asian pigs were clustered into a clade without outgroups (Fig. 2f), suggesting the Hap2 emergence before the divergence of European and Asian pigs. The absence of Hap2 in Sumatra wild boar, Visayan warty pig, Javan warty pig, common warthog, and Pygmy hogs suggests that Hap2 may originate from an extinct species by introgression event or occur at a too low frequency in other species to be detected by the small sample sizes used. Hap1 haplotypes clustered with Sumatra wild boar, Visayan warty pig, and Javan warty pig, indicating the emergence of Hap1 before the divergence of the *Sus* species.

## Phenotypic associations of balancing selection gene *MUC13*

To further investigate the traits under balancing selection of *MUC13*, we explored the phenotypic difference between Hap1 and Hap2 in an F2 population ($N = 1020$) generated by White Duroc boars and Erhualian sows. There was no artificial selection for any traits during the breeding process[49]. The principal component analysis showed that no population stratification existed in the F2 population[16] (Supplementary Fig. 12). We constructed haplotypes using genotypic data genotyped by PorcineSNP60 Genotyping BeadChip 60k of the F2 population and imputed missing variants based on our haplotype reference panel. The haplotypes of *MUC13* were used to associate with more than 300 phenotypes (pertaining to reproduction, meat quality, hematology, cell, growth, and immune) in the F2 population (Supplementary Fig. 13, Supplementary Data 4). Among which one immune- (Enterotoxigenic *Escherichia coli* F4ac, ETEC F4ac) and 8 growth-related traits were significantly different between Hap1 and Hap2 after Bonferroni correction with adjusted $P$ value < 0.05 (Table 1). Individuals with Hap1 have greater risk of the susceptibility to ETEC F4ac than Hap2 (Odds ratio=3.75, Pearson's Chi-squared test $P = 4.1 \times 10^{-35}$), while these 8 growth-related traits have a higher value in the Hap1 group (Student's test $P = 9.1 \times 10^{-5} \sim 1.0 \times 10^{-8}$). The susceptibility towards ETEC F4ac is determined by whether the bacterium adheres to the intestinal receptor or not[4]. In our previous study, *MUC13* was identified as a causative gene affecting the susceptibility to ETEC F4ac in pigs[4], which is a major determinant of diarrhea and mortality in neonatal and young pigs. In this sense, the susceptibility towards ETEC F4ac is one of the selecting pressures on *MUC13*. Another study also demonstrated that *MUC13* affects growth-related traits (rump width and chest width) of pigs[50], consistent with the results of phenotypic

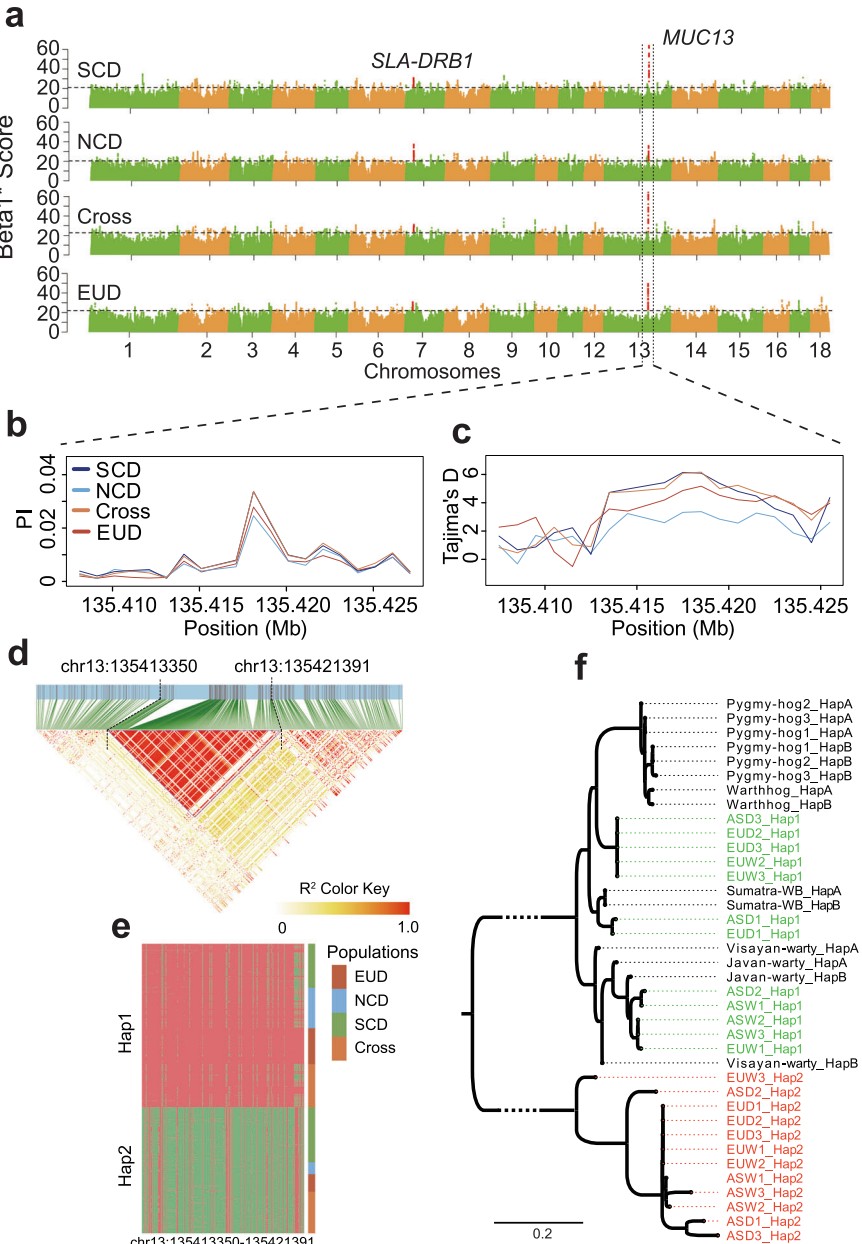

**Fig. 2 | Detecting *MUC13* as a long-term balancing selection gene. a** Scanning the genome for balancing selection loci on autosomes in the following four populations: SCD, Southeastern Chinese domestic pigs; NCD, Northwestern Chinese domestic pigs; Cross, European and Eurasian crossbred pigs; EUD, European domestic pigs. The x-axis shows chromosomes. The y-axis denotes a summary statistic, Beta1* Score, which detects these clusters of alleles within 1 kb at similar frequencies. Each dot denotes a variant. The significance threshold was set to a false discovery rate of 0.01%. The nucleotide diversity **b** and Tajima's D **c** at *MUC13* locus by a sliding 1 kb window. Colored lines represent populations. **d** The Linkage Disequilibrium (LD) pattern at *MUC13* locus. The color of the grid represents the degree of LD measured by R². **e** Two haplotype patterns inside the haplotype block of *MUC13* in populations EUD, NCD, SCD, and Cross. Rows represent haplotypes sorted by type and population. Columns denote variants from position chr13:135413350 to 135421391. The lattices in red/green indicate reference/alternative alleles. **f** The phylogenetic relationship of haplotypes within the balancing selection block from three Asian domestic pigs (ASD), three Asian wild pigs (ASW), three European domestic pigs (EUD), three European wild pigs (EUW), one Indonesian wild boar from Sumatra (Sumatra-WB), one Visayan warty pig from the Philippines (Visayan-warty), one Javan warty pig from Indonesia (Javan-warty), one common warthog from Africa (Warthhog), and three Pygmy hogs from India (Pygmy-hog). The Pygmy-hog2_HapA represents the one of haplotypes of the Pygmy-hog2 individual. The ASD3_Hap1 represents haplotype 1 of the ASD3 individual.

associations. Collectively, the Hap1 confers diminished resistance to ETEC F4ac but is positively associated with growth-related traits.

## Identification of genes under positive selection related to stature

The body size of various pig breeds varies substantially. We herein employed large stature pigs (pDLY: Duroc and Large white and Landrace and their crosses, N = 164; EHL: Erhualian, N = 132) and small stature pigs (BMX: Bamaxiang, N = 84; TBT: Gansu and Sichuan and Tibetan and Yunnan Tibetan, N = 120) to detect candidate genes controlling body size[51]. We determined the $F_{ST}$ value across four pairwise designs (TBT vs. EHL; TBT vs. pDLY; BMX vs. EHL; BMX vs. pDLY), and the intersection of their top 1% variants revealed 139 variants under positive selection (Fig. 3a, Supplementary Data 5). After annotating these variants, we found that 36 candidate genes overlap with them. Of which genes *PLAG1* and *XKR4* had previously been identified

**Table 1 | The phenotypic difference between haplotypes of *MUC13***

| Trait | Hap1 | | Hap2 | | *P* Value | Additive effect | Dominance effect |
|---|---|---|---|---|---|---|---|
| | *N* | Incidence Or Mean ± SD | *N* | Incidence Or Mean ± SD | | | |
| **ETEC F4ac adhesion** | 711 | 0.67 | 815 | 0.35 | $4.1 \times 10^{-35}$ (OR = 3.75) | 0.328*** | 0.130*** |
| **Carcass straight length (cm)** | 854 | 97.04 ± 6.87 | 1000 | 95.31 ± 7.52 | $2.3 \times 10^{-07}$ | 1.823*** | 0.255 |
| **Carcass diagonal length (cm)** | 854 | 80.71 ± 5.89 | 1000 | 79.39 ± 6.41 | $3.9 \times 10^{-06}$ | 1.384*** | 0.249 |
| **Small intestine length (m)** | 856 | 15.88 ± 1.93 | 1002 | 15.46 ± 2.33 | $3.2 \times 10^{-05}$ | 0.429*** | 0.075 |
| **Fourth cervical vertebra length (cm)** | 854 | 2.03 ± 0.20 | 1002 | 1.98 ± 0.20 | $2.2 \times 10^{-06}$ | 0.048*** | 0.001 |
| **Fifth cervical vertebra length (cm)** | 853 | 1.99 ± 0.19 | 997 | 1.95 ± 0.18 | $9.2 \times 10^{-07}$ | 0.045*** | 0.007 |
| **Sixth cervical vertebra length (cm)** | 851 | 2.07 ± 0.20 | 995 | 2.03 ± 0.20 | $4.9 \times 10^{-05}$ | 0.040*** | 0.006 |
| Seventh cervical vertebra length (cm) | 851 | 2.29 ± 0.21 | 999 | 2.23 ± 0.21 | $1.0 \times 10^{-08}$ | 0.059*** | 0.004 |
| Neck Bone Length (cm) | 853 | 17.32 ± 1.41 | 1003 | 17.06 ± 1.49 | $9.1 \times 10^{-05}$ | 0.272*** | 0.109 |

*N*, sample size; ETEC F4ac adhesion, susceptible to Enterotoxigenic *Escherichia coli* F4ac; OR, Odds ratio; Neck Bone Length, total length of cervical vertebra; Hap1 or Hap2, haplotypes of *MUC13*. The significance (*P* Value) of ETEC F4ac phenotypic difference between Hap1 and Hap2 was calculated by Pearson's Chi Squared test. The significances (*P* Value) of remaining phenotypic differences between Hap1 and Hap2 was calculated by Student's t test (two-sided). Reported *P* Values are nominal (i.e. not corrected for multiple testing). Proxy SNP 13_135417839, its allele A (G) in complete linkage disequilibrium with the haplotype Hap1 (Hap2) of *MUC13*, was used to estimate the additive and dominance effects with model $Y = B + A*X1 + D*X2$. *Y* denotes the phenotypic value as 0 (resistance to ETEC F4ac) or 1 (susceptible to ETEC F4ac). *B* denotes intercept. *X1* represents the genotypes (A/A, A/G, G/G) of SNP 13_135417839. A/A, A/G, and G/G were recoded as 2, 1, and 0. Coefficient *A* represents the estimated additive effect. To account for dominance, a new variable was constructed, *X2*, with values of 0, 1, and 0 for, respectively, genotypes A/A, A/G, and G/G. Coefficient *D* represents the estimated dominance effect. *** represents that the *P* Value is highly significant. The *P* Values for the additive effects of ETEC F4ac, carcass straight length, carcass diagonal length, small intestine length, fourth cervical vertebra length, fifth cervical vertebra length, sixth cervical vertebra length, seventh cervical vertebra length, and neck bone length are $5.9 \times 10^{-40}$, $1.9 \times 10^{-07}$, $3.7 \times 10^{-06}$, $3.6 \times 10^{-05}$, $1.2 \times 10^{-06}$, $5.9 \times 10^{-07}$, $4.2 \times 10^{-05}$, $5.6 \times 10^{-09}$, and $1.2 \times 10^{-04}$, respectively. The *P* Value for the dominance effect of ETEC F4ac is $5.2 \times 10^{-05}$.

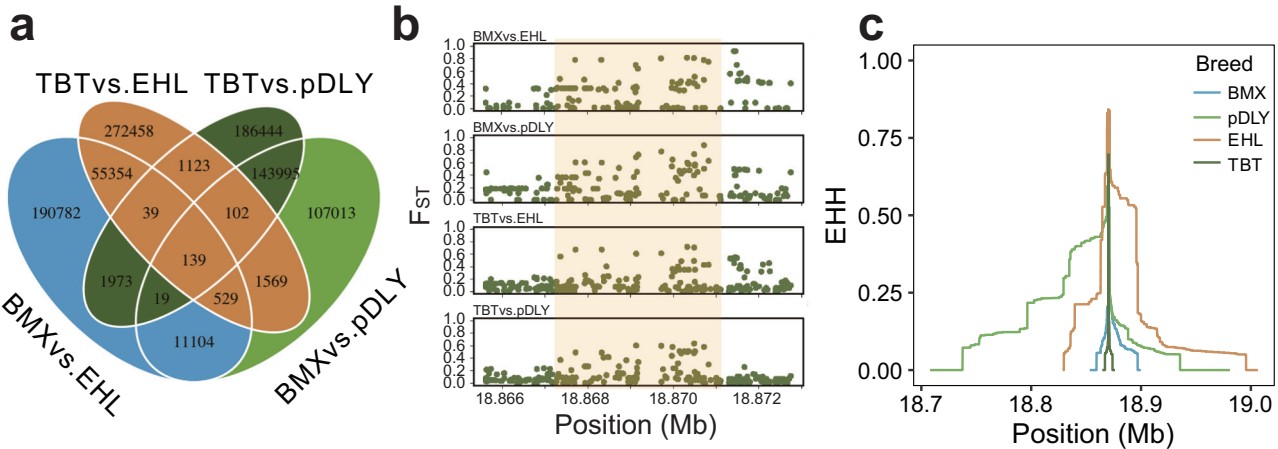

**Fig. 3 | Identification of genes with signatures of positive selection. a** The number of variants with top 1% $F_{ST}$ in each pairwise design (TBT vs. EHL; TBT vs. pDLY; BMX vs. EHL; BMX vs. pDLY) and their intersection. pDLY, Duroc and Large white and Landrace and their crosses; EHL, Erhualian; BMX, Bamaxiang; TBT, Gansu Tibetan and Sichuan Tibetan and Tibetan Tibetan and Yunnan Tibetan. **b** The regional plot for $F_{ST}$ at *NPR3* locus. Each dot represents a variant. The shading indicates a clear signature of positive selection. **c** The EHHs (extended haplotype homozygosity) are calculated based on a proxy variant 16_18869137 in the above four populations. Colored lines represent breeds.

to affect pig body size[16,52]; genes *ZNF521*, *ATP6V1H*, *CCSER1*, and *FAM184B* were shown to influence the stature of cattle, sheep, or horses[53–57]. Notably, one of the candidate genes, *NPR3*, is not only associated with human height and skeletal frame size but also regulates mouse skeletal overgrowth[58–60]. However, *NPR3* has not been reported to be associated with pig body size yet. We examined the regional $F_{ST}$ plot of the NPR3 locus across four pairwise designs. A locus residing in the first intron of *NPR3* exhibited a clear signature of positive selection (Fig. 3b, Supplementary Data 5). Furthermore, we estimated the cross-population extended haplotype homozygosity (XP-EHH) statistic for a proxy variant (16_18869137) across four pairwise designs. All XP-EHHs (TBT vs. EHL, -3.05, $P = 2.32 \times 10^{-03}$; TBT vs. pDLY, -2.84, $P = 4.47 \times 10^{-03}$; BMX vs. EHL, -2.04, $P = 2.46 \times 10^{-03}$; BMX vs. pDLY, -2.56, $P = 1.04 \times 10^{-02}$) also show evidence of positive selection at

a significance level of 0.05. High EHHs of haplotypes could provide evidence of positive selection on this locus[61]. Remarkably, the EHHs were higher in pDLY and EHL than BMX and TBT (Fig. 3c), indicating positive selection on the locus in large stature pigs.

## Discussion

In the current study, we uncovered 41,964,356 autosomal genetic variants for 1,096 individuals in 43 breeds with a sequencing coverage of ~17.1 ×. These variants were applied to construct a high-quality pig haplotype reference panel with accurate genotyping and haplotype phasing. The constructed haplotype reference panel has been shown to be a valuable resource for imputation and deciphering the genetic mechanism of important complex traits in pigs. By reanalyzing a previously published chip-based GWAS study of IMF, our haplotype

reference panel greatly boosted the GWAS power and the accuracy of EBV, indicating the potential use of haplotype panels in the rediscovery of the previously genotyped low-density SNP array data, in line with the previous study[39]. We have shown that these genetic resources can also be used in selection-signature detection and genetic structure analyses. We identified *MUC13* as a long-term balancing selection gene by genomic traces and phenotypic analyses and estimated their haplotypes emergence time. Moreover, positive selection analyses highlighted several genes affecting pig stature.

The high-quality haplotype reference panel depends on two aspects: accurate variant genotyping and high-quality haplotype phasing. Our study shows an accurate variant genotyping with an accuracy of 98.56%, validated by both array replication and mendelian distortion analysis. To maximize the utility of linked variants information, we integrated haplotype-informative sequencing reads, linkage phases within pedigrees, and population LD information to generate high-quality haplotype phases. The integrated information guarantees a high-quality haplotype reference panel. Compared with a previously reported public database Animal-ImputedDB (http://gong_lab.hzau.edu.cn/Animal_ImputeDB/#!/imputation_pig) for online imputation, including 28 pig breeds and 233 individuals worldwide, our reference haplotype panel largely increased the imputed genetic variants accuracy (Mean $R^2$ from 0.82 to 0.89) and the number of variants (from 4,792,133 to 41,964,356) when using randomly selected 50k target variants[8]. Generally, the imputed accuracy of rare variants (MAF ≤ 0.05) is low compared with common variants[37,38]. Remarkably, the accuracy of imputed rare variants in our reference haplotype panel is up to 0.83-0.92 using randomly selected 50-300k target variants, exhibiting similar performance to the human haplotype reference panel (UK10K or Haplotype Reference Consortium)[37,38].

IMF is a major determinant of pork quality, and the phenotype is difficult to acquire on a large scale. With our constructed haplotype reference panel, we reanalyzed previously published low-density chip data (Porcine SNP50 Beadchip) and identified five novel loci associated with IMF. The reason why significant loci could not be identified in the chip GWAS is that GWAS statistical power is reduced due to a low Linkage Disequilibrium (LD) between SNPs and potential causative variants (Supplementary Fig. 14)[62]. However, the number of variants increased after genotypes imputation, which strengthens the LD between imputed WGS variants and potential causative variants (Supplementary Fig. 14), resulting in an increased GWAS statistical power. Two candidate genes within novel loci are functionally related to lipogenesis. *RASGRP3* participated in intramuscular fat formation in Japanese Black cattle[63], and *GDF-3* is an adipogenic cytokine under high fat dietary conditions and highly expressed in adipose tissue[64,65]. We suggest that both genes can be considered as novel candidates affecting pig IMF for further validation. Previous studies show that the more unassociated variants with traits used in the construction of genomic feature-based relationship matrix would diminish the prediction accuracy of breeding value[11,40–42]. Xiang et al. pre-selected ~50k variants at bovine genome and developed a XT-50K genotyping array by integrating the functional, evolutionary and pleiotropic information of variants using GWAS, variant clustering and Bayesian mixture models[43]. The relative increase of genomic prediction accuracy for the XT-50K from the standard-50K ranged from 1.4% to 90%, with a median of 14%. Utilizing all variants to construct genomic feature-based relationship matrix and estimate breeding value, we obtained a similar accuracy of EBV of IMF by array genotypes ($R^2$ = 0.50) and imputed genotypes ($R^2$ = 0.51). However, using pre-selected significant variants from GWAS ($P<0.01$), the accuracy of EBV markedly increased by 36.7% based on imputed genotypes compared with array genotypes. Likewise, the phenotypic prediction ability by EBV also improves significantly by 20% using a cvBLUP method using imputed genotypes. In this sense, it would be more effective at predicting EBV using trait-related variants that have been pre-selected. There are some reasons why such a high increase in genomic prediction accuracy in the study: 1) Genotype imputation increases the number of variants in Linkage Disequilibrium with causal mutation or even the number of causal variants, resulting in identifying more novel QTLs. 2) Preselected variants by GWAS mitigate the effects of unrelated variants on genomic prediction. 3) The XT-50K array could be used for genomic prediction across multiple breeds, sexes, and traits, while the preselected variants for IMF were derived from GWAS of a single breed, sex, and trait. Except for pleiotropic variants, they probably harbor specific variants for the single breed, sex, and trait. Although these variants may not suit other traits or breeds in genomic prediction, they have a good performance on the target breed and trait.

To our knowledge, *MUC13* is firstly reported as a long-term balancing selection gene in our study. Our results suggested that it undergoes selection by diarrhea and other unknown traits. The selection parameter indicated by the Beta1* score in the *MUC13* locus was significantly higher than the remaining genome region, with 63.8 in Southeastern Chinese domestic pigs, 35.8 in Northwestern Chinese domestic pigs, 64.3 in European and Eurasian crossbred pigs, and 49.4 in European domestic pigs. In addition, the selection parameter also identified the MHC region (a common balancing-selection region in the vertebrates) under balancing selection, showing its robust detection ability[23,46]. We further demonstrated *MUC13*'s balancing selection by nucleotide diversity (PI) and Tajimas'D. The PI was increased compared with the surrounding region, and the selection region showed a positive Tajimas'D. The above three results confirmed that the locus is under balancing selection. The LD analysis of the selection region identified a strong LD block of ~8kb, including 240 variants located in the first two introns of *MUC13*. We found only two haplotypes (Hap1 or Hap2) in the populations used for scanning long-term balancing selection genes (SCD, NCD, Cross, EUD). The frequency of Hap1 stays at an intermediate level (0.58), again consistent with the characteristics of balancing selection. *MUC13* exhibits a strong signature of balancing selection shared by Chinese and European pigs. Furthermore, we traced the emergence time of these two haplotypes, which suggested that the Hap1 haplogroups were shared for a long time in Suidae animals[20], and the Hap2 might be introgressed from an extinct species to *Sus scrofa* during the formation of *Sus scrofa* or occur at a low frequency in other species. Our previous study demonstrated that *MUC13* is a single copy gene that encodes two transcripts (MUC13A and MUC13B) with distinct PTS domains[4]. The expression of MUC13B in the small intestine can confer enhanced susceptibility to ETEC F4ac, resulting in diarrhea and mortality in neonatal and young pigs. There are two common mechanisms that cause balancing selection: 1) Overdominance occurs, controlled by one trait, where the fitness of the heterozygote is superior to either of the homozygotes. 2) Pleiotropy, multiple traits are governed by a single locus, gives the heterozygote the highest fitness. To inspect whether overdominance occurred in ETEC F4ac, we selected a proxy SNP 13_135417839, in complete linkage disequilibrium with the haplotype of *MUC13*, to estimate the overdominance of ETEC F4ac. The incidence of ETEC F4ac for genotypes AA, AG, and GG of SNP 13_135417839 are 0.78, 0.58, and 0.13, respectively, indicating that it is not the case of overdominance. Thus, we speculated the *MUC13* pleiotropy might cause this balancing selection. Our further analyses indeed show that *MUC13* associates with ETEC F4ac and 8 growth-related traits, confirming its pleiotropy. We then investigate how *MUC13* might affect the growth-related traits. As far as we know, the small intestine at which ETEC F4ac interacts exerts an effect on nutrient absorption in mammals[66], hence may potentially benefit growth-related traits in pigs owing to better nutrient absorption. A previous study also identified *MUC13* as a candidate gene associated with rump width and chest width in pigs[50]. The above results suggested that *MUC13* may play a role in nutrient absorption. Nonetheless, experimental validation is required to elucidate the molecular mechanism further.

Using the $F_{ST}$ method across four pairwise designs, our positive selection analyses identified 36 candidate genes associated with the stature of pigs. Of which six genes have been previously confirmed to affect the stature-related traits of livestock, such as limb bone lengths, rear legs side view, chest girth, hock circumference, body length, body weight, and body size[16,52–57,67]. The skeletal frame is a major determinant of stature. According to previous reports, *NPR3* is associated with human height, human skeletal frame size, and mouse skeletal overgrowth[58–60]. Moreover, we checked the alleles at proxy variant 16_18869137 in the pDLY, EHL, BMX, and TBT populations. The large stature pigs (pDLY, EHL) share the same the major allele, supporting its positive selection.

Taken together, the study provides rich genetic resources from 1,096 pigs with 43 breeds and detected 41,964,356 autosomal variants. With these genetic resources, we construct a high-quality pig haplotype reference panel and demonstrate its applications in imputation and deciphering genetic mechanisms and breeding programs. Furthermore, we deciphered a long-term balancing selection gene *MUC13* in pigs. Positive-selection analyses highlighted *NPR3* as a candidate gene affecting the stature of pigs. The study provided a haplotype resource and insights into the genetic deciphering of complex traits in pigs.

## Methods

### Data generation and variant discovery

The ethics committee of Jiangxi Agricultural University specifically approved this study. All animal work was conducted according to the guidelines for the care and use of experimental animals established by the Ministry of Agriculture of China. The project was also approved by Animal Care and Use Committee (ACUC) in Jiangxi Agricultural University. We collected 950 pig samples (889 non-redundant individuals: 740 sequenced in this study and 149 published in our previous studies) representing 30 breeds for whole-genome sequencing and retrieved 207 resequencing data from the NCBI SRA database (Supplementary Data 1)[26,29–32]. Genomic DNA was extracted from the ear punches or muscles of pigs using a standard phenol-chloroform extraction protocol. The DNA for each sample was sheared into fragments of 200-800 bp (or 300-400 bp). Next-generation genome sequencing libraries were constructed following the standard protocol of the library preparation kit. Genome sequencing was performed to generate 150 bp (or 125 bp, 100 bp) paired-end reads on the Illumina HiSeq or MGISEQ-2000 platform according to the manufacturer's standard protocols. Using the Fastp v0.20.0[68], we removed reads with ≥ 10% missing ("N") bases or quality score ≤ 20 for ≥ 50% of bases. All clean reads were aligned to the Sus scrofa reference genome 11.1[69] using BWA v0.7.17[70] (BWA-MEM algorithm, default parameters). The sequencing depth and coverage were calculated by Mosdepth v0.3.2[71]. We sorted the bam files of mapped reads by genome position and marked PCR duplication using Picard v2.21.4 (http://broadinstitute.github.io/picard). The gvcf files were generated using the official pipeline of GATK v4.1.7.0, including program BaseRecalibrator, ApplyBQSR, and HaplotypeCaller. We then jointly genotyped all gvcf files to obtain the final vcf file using GATK GenotypeGVCFs. The variants were filtered with the following criteria: (1) SNP: QD < 2.0, QUAL < 30.0, MQ < 40.0, SOR > 3.0, FS > 60.0, MQRankSum < -12.5, ReadPosRankSum < -8.0; (2) INDEL: QD < 2.0, QUAL < 30.0, MQ < 40.0, FS > 200.0, ReadPosRankSum < -20.0. In addition, variants with reads coverage depth beyond the whole-genome average depth of 1.5 times were removed. Due to the low quality of SNP near INDEL, we used 325 individual array genotypes to evaluate the accuracy of SNPs that are within 1-50 bp of Indels (Supplementary Fig. 15). The accuracy of SNPs reaches saturation at a distance of 5bp from Indels, we thus removed the SNP within 4 bp of INDEL using bcftools v1.9[72]. To reduce the genotyping error in low-depth sequencing samples, the genotypes of variants with less than six reads coverage were assigned to missing for

each sample. Based on all individuals and variants, we filtered variants with the following criteria: Minor allele count ≤ 2, Missing rate ≥ 0.05, and samples with call rate ≤ 0.85. The IBS was calculated using Plink v1.9 with parameter "--genome"[73]. Finally, 937 high-quality samples, 41,326,535 SNPs and 3,864,622 INDELs were obtained. Of these samples, there are 44 trios and 135 duos; 137 individuals were genotyped with the Illumina PorcineSNP50 BeadChip; 188 individuals were genotyped with the Illumina PorcineSNP60 BeadChip.

### Population structure analyses

All analyses were based on autosomal variants. The genetic distance between individual pairs was calculated using Plink v1.9 with parameter "--distance-matrix"[73], and a genetic distance matrix was obtained. The matrix was used to build Neighbor-Joining tree using the Neighbor program in PHYLIP v3.69 (https://evolution.genetics.washington.edu/phylip.html). We visualized the Neighbor-Joining tree with the FigTree v1.4.4 software (http://tree.bio.ed.ac.uk/software/figtree).

### Construction of the haplotype reference panel

Firstly, we established haplotype blocks based on sequencing reads using HapCUT2 v1.3.1 with parameters "--mbq 13 --mmq 20 --indels 1"[33] for each individual. Vcf file A with haplotype blocks was obtained. Secondly, based on the pedigree information (Supplementary Data 1), we constructed haplotype scaffolds using LINKPHASE3 v1.0 included in the PHASEBOOK package with parameters "HALFSIB_PHASING yes HMM_PHASING yes CHECK_PREPHASING yes"[34]. Vcf file B with haplotype scaffolds was generated. Finally, the haplotype reference panel was constructed by combining haplotype blocks (vcf A) and scaffolds (vcf B) with population LD information using SHAPEIT4 v4.4.2 with parameters "--sequencing --scaffold --use-PS 0.0001"[35].

### Imputation, GWAS, and breeding value estimation

Based on the genotypic data genotyped by Porcine SNP50 Beadchip, the haplotype construction was performed using SHAPEIT4 v4.4.2 with default parameters[35]. We then conducted genotype imputation using IMPUTE5 v1.5.5 with default parameters[74]. The Porcine SNP50 Beadchip genotypes for 1,490 Duroc pigs and their IMF content phenotype were downloaded from a published GWAS study[2]. GWASs were performed using the Genome-wide Efficient Mixed-Model Association method (GEMMA v0.98, with default parameters) with a univariate linear mixed model[75]. The estimated breeding value was calculated using gcta v1.93.2Beta[76]. We first generated a genetic relationship matrix based on pre-selected variants ($P < 0.01$ in GWAS) using the parameter "--make-grm". We then used BLUP (best linear unbiased prediction) method to estimate the breeding value using parameters "--grm --pheno --reml-pred-rand" or "--reml --grm --pheno --cvblup" (leave-one-individual-out BLUP analysis).

### Balancing selection analyses

To detect the genome traces of balancing selection, we used BetaScan2 v1.0 to calculate Beta1* Score with parameters "-w 1000 -fold"[77]. The nucleotide diversity (PI) and Tajimas'D were calculated by a sliding 1kb window using VCFtools v0.1.13[78]. We employed LDBlockShow v1.40 to generate and visualize the LD pattern of haplotype block with parameter "-SeleVar 2". The multiple sequence alignment for the haplotype data was conducted using Clustalo v1.2.4[79], and the maximum-likelihood tree was built using Fasttree v2.1.11 (http://www.microbesonline.org/fasttree)[80]. We annotated variants using the online Ensembl Variant Effect Predictor (VEP, https://asia.ensembl.org/info/docs/tools/vep/index.html). The F2 population used for phenotypic analyses was constructed with two founder breeds: White Duroc and Chinese Erhualian. Briefly, two White Duroc boars were mated to 17 Erhualian sows, and 9 F1 boars were then intercrossed with 59 F1 sows avoiding full-sib mating to generate 1912 F2 animals[49]. A microscopic enterocyte adhesion assay was used to record in vitro ETEC F4ac

adhesion phenotypes[4]. Animals at day $240 \pm 3$ were slaughtered for recording Carcass straight length, Carcass diagonal length, Carcass weight, Small intestine length, Neck Bone length, Anterior brachial bone length, and Ear area.

## Positive selection analyses

Four pairwise designs (TBT vs. EHL; TBT vs. pDLY; BMX vs. EHL; BMX vs. pDLY) were used for identifying candidate variants under positive selection on body size. The Weir and Cockerham weighted $F_{ST}$ values of variants were calculated using software VCFtools v0.1.13 with parameter "--weir-fst-pop"[78]. Briefly, $F_{ST}$ was estimated as follows:

$$F_{ST} = \frac{MSP - MSG}{MSP + (n_c - 1)MSG}$$

*MSG* denotes the observed mean square errors for variants within populations. *MSP* represents the observed mean square errors for variants between populations. $n_c$ is the average sample size after incorporating and correcting for the variance of sample size over population,

$$MSG = \frac{1}{\sum_{i=1}^{s} n_i - 1} \sum_{i}^{s} n_i p_{Ai}(1 - p_{Ai})$$

$$MSP = \frac{1}{s - 1} \sum_{i}^{s} n_i (p_{Ai} - \bar{p}_A)^2$$

$$n_c = \frac{1}{s - 1} \sum_{i=1}^{s} n_i - \frac{\sum_i n_i^2}{\sum_i n_i}$$

Where $n_i$ denotes the sample size in the ith population, $p_{Ai}$ is the frequency of variant allele A in the ith population, and $\bar{p}_A$ is a weighted average of $p_A$ across populations. Moreover, XP-EHH, a cross-population statistic, was used to detect the positive selection further. It is based on extended haplotype homozygosity (EHH) statistics inside each population. We used the rehh v3.2.2 package implemented in R to calculate EHH and XP-EHH statistics[81]. The parameter to consider the positive selection was *P* Value of XP-EHH < 0.05.

## Reporting summary

Further information on research design is available in the Nature Portfolio Reporting Summary linked to this article.

## Data availability

The haplotype data, including genotype data and phasing information, generated in this study have been deposited in the National Genomics Data Center database (NGDC) under accession code GVM000479. The 740 raw resequencing genome datasets generated in this study have been deposited in NGDC/GSA under accession code CRA011506 (https://ngdc.cncb.ac.cn/gsa). It is available under restricted access due to unpublished work involving genome structure variation and evolution; access can be obtained by written request to L.H. (lushenghuang@hotmail.com) who will aim to respond to requests within 2 weeks. The 149 genomes we previously published are publicly available[26,29–32]. The details of all the genomes analyzed in the study, including the 207 publicly available resequencing datasets we retrieved, are included in Supplementary Data 1. The information on the pedigree structure is provided in Supplementary Data 1. The genotypes of individuals and the phenotypic data of IMF from Ding et al.[2] are available at https://figshare.com/s/be3bb2047df324c8a77e. The GWAS summary statistics generated in the study have been deposited in NGDC under accession code GVP000007. The summary statistics for phenotypic associations of gene *MUC13* are provided in Supplementary Data 4.

## Code availability

The codes for variants calling, population structure analyses, construction of the haplotype reference panel, imputation of genotypes, breeding value estimation, balancing selection analyses and positive selection analyses are available from the GitHub repository (https://github.com/xinkaitong/1k-pig-genomes)[82].

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

## Acknowledgements

This work was supported by grants from the National Key R&D Program of China (2022YFF1000102, H.A. hosts the fund), National Natural Science Foundation of China (31790413, L.H. hosts the fund), and Major Scientific and Technological R & D Projects of Jiangxi Provincial Department of Science and Technology (No. 20213AAF01010, L.H. hosts the fund).

## Author contributions

L.H. conceived and designed the study, directed the project, provided all data and computational resources, supervised bioinformatic and statistical analyses, and revised the paper. Z.Z. and H.A. codesigned the study, supervised bioinformatic and statistical analyses, and revised the paper. X.T. collected the resequencing data and performed most of the analyses and wrote the paper. D.C., J.H., and S.L. collected the resequencing data and performed part of analyses. Z.L. revised the paper.

## Competing interests

The authors declare no competing interests.
