## [Peer Review File · Nature Communications]

Accurate haplotype construction and detection of selection signatures enabled by high quality pig genome sequencesREVIEWER COMMENTS

Reviewer #1 (Remarks to the Author):

In this paper, Tong et al presents whole-genome resequencing data for almost 900 pigs and use these data to construct genome-wide haplotype information. They then show the power of this resource for imputation and genetic analysis in the pig. This is a valuable contribution to pig genome research and the results presented are promising.

Specific comments

1. Population structure (L147-9). I don't think this section is an advance compared with the many, many previous papers on genetic relationships among pig populations. The analysis is based on more data than used in previous work but the outcome is the same as far as I can see and I don't think this analysis is of broad interest. I also find it confusing to include crossbred individuals in this analysis. Of course, these populations will be intermediate between the ancestral populations. I suggest that this section is deleted unless the authors can highlight what is a new insight of broad interest appropriate for this journal. I think the paper contains sufficient information without this section.

2. GWAS. The improvement in GWAS analysis is impressive, almost too good to be true. It would have been nice to validate some of the top SNPs using genotyping but that is perhaps not possible if it is data from another group.

3. Balancing selection. The data presented concerning MUC13 identify two 8 kb highly divergent haplotypes. Please spell out the nucleotide sequence divergence between the two haplotypes, Fig. 3b indicates that it should be about 3.0% which is impressive compared with the 1.2% average sequence divergence between human and chimpanzee haplotypes. The authors suggest that this haplotype has been introgressed from another species but you cannot exclude the possibility that this is an old polymorphism that has been lost in the other species included here or occur at a too low frequency to be detected by the small sample sizes used.

4. MUC13 and NPR3 validation. If you have measured 300 phenotypes in this F2 population, why restrict your analysis to the traits that already have been established in previous studies of MUC13, a negative result for other traits is also an important result to guide future work.

I don't find the validation of the phenotypic effect for these two loci convincing (except for the ETEC association which is already well established) because you observe these in a divergent intercross (F2 population) with poor map resolution. Thus, the significances for various carcass traits you report here may be caused by linked QTLs. You need to back up the results presented in the Table by QTL graphs for the actual chromosomes to show that the peak of the QTL signal match MUC13 and NPR3. (A GWAS in an outbred population segregating for these variants would give better map resolution). Thus, I disagree with the statement on line 421-422 that the association was validated by phenotypic association analysis.

Furthermore, I recommend that you report estimates of the additive and dominance effects at these loci. Minor comment: are these P-values corrected for multiple testing? and it is sufficient to present P-values with a single decimal place.

5. I did not find any statement where these haplotype data are made public which I assume is a prerequisite for publication.

Minor comments:

Line 27 and 180: I think it is more appropriate to refer to these as 1874 haploid genomes.

Line 79, Here you state that you sequenced representatives of most of the breeds (N=30) while on line 60 you state that there are possible more than 700 breeds.

Line 237-239. This text is essentially repeated on line 456-458. I suggest it is deleted here as it is better suited for the Discussion.

Line 471-472. The statement is unclear. Do you mean that it has been reported previously as a long-term balancing gene if so, add reference, or, if you mean that this is the first paper indicating long-term selection, please revise the text.

Discussion. There is some redundancy in the Discussion concerning the results on MUC13 and NPR3 which are discussed/summarized in three paragraphs of the Discussion and (in the Abstract). These overlaps should be eliminated.

Reviewer #2 (Remarks to the Author):

Tong et al. used WGS data of 1096 pigs 43 breeds to develop a panel of haplotypes for imputation; carried out GWAS for intramuscular fat content and study a few gene in evolutionary contexts. Overall the manuscript is easy to follow. The standard methodologies commonly employed for such research were used.

L108: After quality control, eight breeds had 1-2 samples. Is not this sample size too small to include a breed in the analysis?

Fig.2a. If I understand correctly, the plots are average of 10 imputed animals, only one round. The imputation should be repeated for large number of rounds including breeds which are not 'common'. Though it is important that the imputation accuracy is high for (common) commercial pigs, but generally, those have good reference populations. It is important to know the performance for breeds, which may not have good references.

Fig.2b: Large increase in $-\log_{10}(p)$ with imputed WGS variants. Any explanation why some regions showed high increase in significance level while no changes in many others (ex. Chr2, 7, distal end of 9).

Fig.2c: Were the IMF phenotypes corrected for systematic fixed and random effects? The X-axis label should be 'IMF phenotype'

L248-250: "The correlation (R^2) between phenotypic value and imputation EBV ($R^2 = 0.67$) increased by 36.7% compared with chip EBV ($R^2 = 0.49$)". This is a remarkable improvement compared to several studies in livestock where sequence variants were included to increase genomic prediction accuracy (e.g. Xiang et al. 2021 Nat Commun., 10.1038/s41467-021-21001-0). Could you discuss the reasons to see such a high increase?

L254: imputed EBV or selected imputed WGS variants?

L 266: 'outstanding' -> distinctive

L321: Validation of balancing selection gene MUC13 – I wonder if this was the right design to validate. "We found that only two 334 haplotypes (Hap1 or Hap2) existed in this balancing selection block region in the F2 population". Were there only 2 haplotypes in the parental population used to create F1 and what were the frequencies of haplotypes? What is the expectation of frequency changes in a 'non-selected' population in 2 generations?

Table 1. ETEC F4ac – Please specify '+' and '-'. what test did you do (McNemar's test)? Odds ratio may be informative.

Fig.4e: I do not understand this figure; more details required.

L556: Minor allele count ≤ 2 ; is it not too liberal?

Reviewer #3 (Remarks to the Author):

In their manuscript Tong et al. describe the development of a genetic resource for pigs, that will provide higher power and resolution in GWAS and genomic Selection. The strength of the new resource is shown for the identification of potential genes involved in quantitative traits (e.g. IMF) and/or under selection (MUC13 and NPR3). The manuscript in general is written clearly, although there are some sections where the English language can be improved (see minor comments for details).

The literature cited seems to be somewhat biased and it gives the impression that this might have been done to strengthen the “innovativeness of the manuscript”. E.g. in the introduction (lines 74-75) it is stated that it is not known whether pig-specific balancing selection genes exist, while a publication in PLoS genetics clearly showed balancing selection for a 200 Kb deletion affecting the BMPER and PPS9 genes in pigs (Derks et al. PLoS gen. 2018, 14: e1007661). Another example is a reference to unpublished results of the authors citing different evolutionary time points than what is generally accepted and supported by many previous other studies. The authors use 219 Kya for the last common ancestor of European and Asian wild boar, while previous studies estimate this to be ~ 1 Mya. A similar discrepancy is seen for the divergence of the Sus species which the authors claim to be 1.36 Mya while multiple other studies estimate this to be ~3.5-4 Mya. The latter estimates are not only derived from genetic studies but are also supported by fossil records.

The same for the results and discussion about the improved resolution and power for GWAS when using imputed SNPs. This has been shown in previous studies (e.g. a recent study for pigs is described by Derks et al 2021, Genomics 113: 2229).

Line 350-351: “but benefits growth-related traits, which supports the hypothesis of balancing selection.” I don’t think this statement is correct. This would be the case if this is observed in a population that is under selection for high growth like is the case for domestic breeds” However, this is not the case for wild boar populations. Why would the Hap2 haplotype be under selection in all wild Sus scrofa populations (and not in the other suids, many of which even are similar in size as wild boar). The same applies for the statement made in lines 385-386.

Lines 484-486: I do not agree with the statement that the observed frequency of 0.58 is consistent with balancing selection. The allele frequencies in the parental populations are already around 50% and the authors state that there was no selection applied in the F1 and the F2 generations. So an observed allele frequency of Hap1 of 0.58 in the F2 population is what you would expect.

Line 552: Removing only SNPs that are within 1 bp of an INDEL seems rather relaxed and still potentially leads to inclusion of false positives.

Minor comments

Lines 78-79: The authors state that the individuals used in their study cover most pig breeds in Asia and parts of the breeds worldwide. This contradicts an earlier statement made in lines 60-61 that there are 730 pig breeds worldwide of which two-third are found in China and Europe.

Line 78: insert “we” before “sequenced”

Line 78: Change “covered” by “covering”

Line 115: delete “is” in “rate is ranged”

Lines 117 and 119: “Mendel error” should be “Mendelian error”

In lines 152-164: Remove “the” before the abbreviations SCT1 and LA. Als change “was” clustered to “clustered. So instead of (line 155) “The STC1 was clustered” write “STC1 clustered”.

I although think it sounds better if the first time an abbreviation is used to first write it in full: E.g. in line 152-153, change “Although the SCT1 (Luding Tibetan pigs, N = 50) and the SCT2 (Litang Tibetan pigs, N = 12) lived in” I suggest to write “Although Luding Tibetan pigs (SCT1, N = 50) and Litang Tibetan pigs (SCT2, N = 12) lived in”

Line 163: Change “The GST (Gansu Tibetan pigs, N = 14) was located” toe “Gansu Tibetan pigs (GST, N = 14) are located”

Line 171: insert “a” before “haplotype”

Line 174: Change “built” to “build”

Line 187” Change “As” to “Because of”

Line 198: “elected” should be “selected”

Lines 231-232: Insert “the” before “chip”

Line 232: Change “Of which three leading” to “Three leading.....”

Line 132: Change “gene” to “genes”

Line 274: Change “Expectedly” to “As expected”

Line 335: Insert “the” before “older”

Line 344: “Compared” instead of “compare”

Line 364: Change “varied” to “varies”

Lines 425-428: There are multiple errors in this sentence. Change this sentence to “Further analyses in various Tibetan pig populations show that the Luding Tibetan pigs (STC1) and the Litang Tibetan pigs (STC2) did not cluster together in the Neighbor-Joining tree and have distinct ancestral compositions, although they live in the same geographical location.”

Lines 429-430: Again remove “the” before STC1.

Line 429: Change “artificial” to “artificially”

Line 449: Replace “their” by “the”

Line 453: Insert “the” before “Porcine”

Lines 457-458: I suggest connecting these two sentences. “.....Black cattle and GDF-3 is”

Line 485: Delete “analytic”

Line 510: Change “under a positive selection” to “is under positive selection”

Line 510: Insert “the” before “major”

Response to Reviewers

January 14, 2023

We carefully checked the comments and revised the paper by point to point. All revisions were highlighted in red in the manuscript with track. The point-by-point responses to the concerns are listed as follows.

Responses to Reviewer #1

In this paper, Tong et al presents whole-genome resequencing data for almost 900 pigs and use these data to construct genome-wide haplotype information. They then show the power of this resource for imputation and genetic analysis in the pig. This is a valuable contribution to pig genome research and the results presented are promising.

Response: We greatly thank Reviewer #1 for these positive assessments.

Specific comments:

1. Population structure (L147-9). I don't think this section is an advance compared with the many, many previous papers on genetic relationships among pig populations. The analysis is based on more data than used in previous work but the outcome is the same as far as I can see and I don't think this analysis is of broad interest. I also find it confusing to include crossbred individuals in this analysis. Of course, these populations will be intermediate between the ancestral populations. I suggest that this section is deleted unless the authors can highlight what is a new insight of broad interest appropriate for this journal. I think the paper contains sufficient information without this section.

Response: In order to focus our other novel results and discoveries, we followed the Reviewer #1 for this suggestion, and deleted this section of *Population structure*, Fig. 1, the related discussion, and the corresponding methods in the revised main text. In order to present geographical distribution (i.e., sampling sites or country of origin of breeds) of the study populations and their genetical classification, we have added a description (main text: **lines 102-107**) into the section of *Description of resequencing data and genome variants discovery* as follows: "These breeds are mainly distributed in China and Europe (Supplementary Fig. 2). All individuals could be geographically divided into seven groups: Asian wild pigs, Southeastern Chinese domestic pigs, Northwestern Chinese domestic pigs, Eurasian crossbred pigs, European crossbred pigs, European domestic pigs, and European wild pigs (Supplementary Fig. 3).", and placed the Fig. 1a (renamed to Supplementary Fig. 2) and the Fig. 1c (renamed to Supplementary Fig. 3) in the revised supplementary file.

2. GWAS. The improvement in GWAS analysis is impressive, almost too good to be true. It would have been nice to validate some of the top SNPs using genotyping but that is perhaps not possible if it is data from another group.

Response: We thank Reviewer #1 for raising this very important comment. The data of genotypes and phenotypes for IMF were downloaded from Yang *et al.* to evaluate the GWAS power and GS accuracy using unrelated population. We have difficulties to further access the DNA for genotyping the top SNPs. But we tried to strengthen the reliability of these novel loci by haplotype analysis using 50k array data. The haplotypes were constructed using ten SNPs surrounding top SNPs at the novel loci in

imputation GWAS. We then analyzed adjusted phenotype (adjusted by 10 PCAs of genotype) variation among different haplotypes using analysis of variance (ANOVA). The results showed that the adjusted phenotype was significantly different (genome-wide threshold, $P < 5 \times 10^{-8}$) among haplotypes at each locus (**Supplementary Table 3**). Besides, we also performed ANOVA for haplotypes at the top locus at each autosome in chip GWAS. The method is the same as above. The results showed that the significance of 16 of 18 loci did not surpass the genome-wide threshold (**Table R1**). The significance ($P_value = 4.6 \times 10^{-08}$) for the locus on chromosome 15 was slightly higher than the genome-wide threshold. The locus on chromosome 3 presented high significance ($P_value = 4.7 \times 10^{-15}$) in ANOVA, and it (3_11528693, rs329147631) has also been reported in the original study¹. Taken together, we thought that these five novel loci in imputation GWAS emerged based on phenotypic variation across their haplotypes, which strengthens the reliability of these signals. The above results have been added to the section of *Improved power of GWAS by the haplotype reference panel* (main text: **lines 206-213**) as follows: "On the other hand, to strengthen the reliability of these novel loci, we performed haplotype analysis using 50k array data. The haplotypes were constructed using ten SNPs surrounding top SNPs at these novel loci. The analysis of variance (ANOVA) showed that the adjusted phenotype (adjusted by 10 PCAs of genotype) was significantly different (genome-wide threshold, $P < 5 \times 10^{-8}$) among haplotypes at each locus (Supplementary Table 3). Herein, we thought these five novel loci emerged based on phenotypic variation across their haplotypes, which strengthens the reliability of these signals."

Supplementary Table 3 Chip-based haplotype analysis for top SNPs at novel loci from imputation GWAS.

Chr	Top SNP	MAF	No.Hap	Begin.Hap	End.Hap	Df	F_value	P_value
3	3_106129949	0.025	25	105,756,806	106,275,479	24	3.88	6.9E-10
5	5_62861562	0.012	29	62,753,754	63,543,086	28	11.11	9.0E-47
8	8_66234703	0.319	25	65,489,709	66,846,245	24	9.87	2.6E-35
9	9_10302512	0.012	12	10,180,053	10,403,658	11	5.87	1.7E-09
13	13_162125842	0.027	18	160,745,027	162,777,523	17	7.95	6.1E-20

Chr, chromosome; MAF, minor allele frequency; No.Hap, the number of haplotype pattern; Begin.Hap, start position of haplotype; End.Hap, end position of haplotype; Df, degree of freedom; F_value, F value in analysis of variance (ANOVA); P_value, P value in ANOVA.

Table R1. Chip-based haplotype analysis for top SNP at each autosome from chip GWAS.

Chr	Top SNP	MAF	No.Hap	Begin.Hap	End.Hap	Df	F_value	P_value
1	1_21060443	0.375	27	20,859,637	21,244,650	26	1.46	6.3E-02
2	2_24096039	0.068	14	23,839,965	24,320,775	13	1.47	1.2E-01
3	3_11528693	0.086	18	11,396,652	11,636,159	17	6.38	4.7E-15
4	4_106919782	0.056	15	106,773,637	107,078,063	14	3.49	1.1E-05
5	5_9698050	0.018	24	9,586,180	9,952,581	23	2.46	1.3E-04
6	6_168268278	0.442	28	167,986,191	168,615,818	27	1.89	3.6E-03
7	7_117427086	0.259	37	117,250,109	117,517,380	36	1.69	6.1E-03
8	8_6548221	0.463	29	6,164,113	6,676,590	28	1.85	4.4E-03
9	9_118528867	0.012	29	118,122,558	118,873,856	28	2.28	1.5E-04
10	10_50442049	0.409	31	50,084,600	50,614,314	30	1.24	1.7E-01
11	11_62934212	0.488	32	62,685,704	63,096,569	31	1.24	1.7E-01
12	12_57258583	0.071	24	56,959,881	57,386,276	23	2.51	8.8E-05
13	13_194840009	0.128	30	194,751,655	195,075,779	29	2.01	1.1E-03
14	14_131949286	0.290	31	131,865,293	132,108,988	30	1.44	5.8E-02
15	15_12296852	0.265	21	12,164,277	12,553,011	20	3.72	4.6E-08
16	16_514821	0.222	18	346,456	743,142	17	1.16	2.9E-01
17	17_45617105	0.208	19	45,340,173	45,795,728	18	1.97	8.7E-03
18	18_53079675	0.441	24	52,898,591	53,397,077	23	1.95	4.5E-03

Chr, chromosome; MAF, minor allele frequency; No.Hap, the number of haplotype pattern; Begin.Hap, start position of haplotype; End.Hap, end position of haplotype; Df, degree of freedom; F_value, F value in analysis of variance (ANOVA); P_value, P value in ANOVA.

3. Balancing selection. The data presented concerning MUC13 identify two 8 kb highly divergent haplotypes. Please spell out the nucleotide sequence divergence between the two haplotypes, Fig. 3b indicates that it should be about 3.0% which is impressive compared with the 1.2% average sequence divergence between human and chimpanzee haplotypes. The authors suggest that this haplotype has been introgressed from another species but you cannot exclude the possibility that this is an old polymorphism that has been lost in the other species included here or occur at a too low frequency to be detected by the small sample sizes used.

Response: We thank Reviewer #1 for this valuable comment. The 1.2% average sequence divergence between human and chimpanzee haplotypes was determined by Jukes-Cantor distance in a previous study². By the same method, we calculated the nucleotide sequence divergence between the two 8kb haplotypes in our reference haplotype panel, with an average sequence divergence of 3.04%. The nucleotide sequence divergence between the two haplotypes has been added to the section of *Identification of genes under long-term balancing selection* in the revised manuscript (main text: **lines 254-256**) as follows: "The nucleotide sequence divergence between the two 8kb haplotypes was calculated by Jukes-Cantor distance, with an average

sequence divergence of 3.04%."

The Hap2 is absent in Sumatra wild boar, Visayan warty pig, Javan warty pig, common warthog, and Pygmy hogs (Fig. 2f). In our original manuscript, we speculated that the Hap2 may originate from an extinct species by introgression event. Although we cannot exclude the possibility that the Hap2 has been lost in these five species, its probability could be estimated as follows:

- 1) Suppose these six species come from an ancestor A, and they evolve independently.
- 2) The Hap2 is under a long-term balance selection in pigs, and we thus estimated the frequency of Hap2 in ancestor A with the assistance of the frequency of Hap2 (0.42) in pigs. Subsequently, we got an estimated frequency of Hap2 in ancestor A, $f_{(\text{Hap2})} = 0.42$.
- 3) If a diploid species evolved from ancestor A, the probability that the Hap2 lost is $P_d = (1 - f_{(\text{Hap2})}) \times (1 - f_{(\text{Hap2})})$.
- 4) Finally, the probability that the Hap2 lost in five species but preserves in one species is $P = C_6^1 \times (P_d)^5 \times (1 - P_d)^1 = 0.017$.

Due to the $P = 0.017 < 0.05$, we think that the Hap2 has been lost in these five species but not in pigs is a small probability event. Indeed, we cannot exclude the possibility that the Hap2 occurs at a too low frequency in other species to be detected by the small sample sizes used. Therefore, we have corrected the corresponding statement in the revised manuscript (main text: lines 268-271) as follows: "The absence of Hap2 in Sumatra wild boar, Visayan warty pig, Javan warty pig, common warthog, and Pygmy hogs suggests that Hap2 may originate from an extinct species (*Sus scrofa*) by introgression event or occur at a too low frequency in other species to be detected by the small sample sizes used."

4. MUC13 and NPR3 validation. If you have measured 300 phenotypes in this F2 population, why restrict your analysis to the traits that already have been established in previous studies of MUC13, a negative result for other traits is also an important result to guide future work. I don't find the validation of the phenotypic effect for these two loci convincing (except for the ETEC association which is already well established) because you observe these in a divergent intercross (F2 population) with poor map resolution. Thus, the significances for various carcass traits you report here may be caused by linked QTLs. You need to back up the results presented in the Table by QTL graphs for the actual chromosomes to show that the peak of the QTL signal match MUC13 and NPR3. (A GWAS in an outbred population segregating for these variants would give better map resolution). Thus, I disagree with the statement on line 421-422 that the association was validated by phenotypic association analysis.

Furthermore, I recommend that you report estimates of the additive and dominance effects at these loci. Minor comment: are these P-values corrected for multiple testing? and it is sufficient to present P-values with a single decimal place.

Response: We took this critical comment from Reviewer #1 and added further studies accordingly. Following the reviewer's suggestion, the 309 phenotypes measured in the F2 population which were divided into 7 categories, reproduction-, meat quality-,

hematology-, cell-, growth-, immune-related traits, and others based on their physiological and biochemical characteristics were further analyzed and presented (**Supplementary Table 5**). Their differences between haplotype Hap1 and Hap2 of *MUC13* were estimated (**Supplementary Fig. 13**). One immune- (ETEC F4ac) and 8 growth-related traits (carcass straight length, carcass diagonal length, small intestine length, fourth cervical vertebra length, fifth cervical vertebra length, sixth cervical vertebra length, seventh cervical vertebra length, total length of cervical vertebra) were significantly different between Hap1 and Hap2 after Bonferroni correction with adjusted P value < 0.05 . Individuals with Hap1 have higher risk of the susceptibility to ETEC than Hap2 (Odd ratio=3.75, Pearson's Chi-squared test $P = 4.1 \times 10^{-35}$), while these 8 growth-related traits have a higher value in the Hap1 group (Student's test $P = 9.1 \times 10^{-5} \sim 1.0 \times 10^{-8}$). The remaining 300 phenotypes have no significant difference between Hap1 and Hap2 after multiple testing correction (details see **Supplementary Table 5**). The results have been added to the section of *Phenotypic associations of balancing selection gene MUC13* (main text: **lines 304-312**) as follows: "The haplotypes of *MUC13* were used to associate with more than 300 phenotypes (pertaining to reproduction, meat quality, hematology, cell, growth, and immune) in the F2 population (Supplementary Fig. 13, Supplementary Table 5). Among which one immune- (Enterotoxigenic *Escherichia coli* F4ac, ETEC F4ac) and 8 growth-related traits were significantly different between Hap1 and Hap2 after Bonferroni correction with adjusted P value < 0.05 (Table 1). Individuals with Hap1 have greater risk of the susceptibility to ETEC F4ac than Hap2 (Odd ratio=3.75, Pearson's Chi-squared test $P = 4.1 \times 10^{-35}$), while these 8 growth-related traits have a higher value in the Hap1 group (Student's test $P = 9.1 \times 10^{-5} \sim 1.0 \times 10^{-8}$).".

We have generated QTL graphs for 10 traits presented in Table 1 in the F2 population (**Fig. R1**). The balancing selection loci overlapped with the peak of ETEC F4ac QTL (**Fig. R1a-b**). Although no QTL was detected at the *MUC13* locus for growth-related traits in the F2 population (**Fig. R1c-k**), we correlated haplotypes of *MUC13* with 309 phenotypes to explore potential traits under selection (**Supplementary Fig. 13**). The *MUC13* is significantly associated with growth-related traits. In addition, a previous study also demonstrated that *MUC13* affects growth-related traits (rump width and chest width) of pigs³. Thus, we speculated that *MUC13* affects other unmeasured phenotypes, such as nutrition absorption ability, in turn contributing to growth-related traits. The effect of *MUC13* on nutrition absorption may be shown partly by growth-related traits, resulting in the inability to the identification of significant QTL nearby *MUC13*. Combining this suggestion and corresponding analyses, we have corrected the section title "Validation of balancing selection gene *MUC13*" to "Phenotypic associations of balancing selection gene *MUC13*". The related statements have been reworked in this section and the section of *Discussion* (main text: **lines 316-320, 437-438, 461-462, 464, 466-468**).

The results showed that no QTL was found at *NPR3* for the four traits presented in Table 1 in the F2 population (**Fig. R1c, e, j, k**). Gene *NPR3* was pinpointed by positive analyses (F_{ST} and XP-EHH) in four pairwise designs involving large and small stature pigs. Besides, *NPR3* is functionally related to stature-related traits (height, skeletal

frame size, skeletal overgrowth) in human and mouse⁴⁻⁶. However, we did not validate *NPR3* in the F2 population due to no QTL being found. The F2 population was generated by two large-stature pigs (White Duroc boars and Erhualian sows), which may lead to no segregation of causative variants controlling *NPR3* for stature. The same phenomenon was also observed in the famous gene *PLAG1* that affects body size⁷, no QTL was detected at *PLAG1* (on chromosome 4) in the F2 population (**Fig. R1c-g**). Thus, we could not exclude the possibility that *NPR3* is responsible for pig stature. Taken together, we consider *NPR3* as a candidate gene for pig stature and deleted the phenotypic associations of *NPR3* in the F2 population, and reworked the statements on *NPR3* validation (main text: **lines 31-32, 485**).

The additive and dominance effects have been estimated in the revised Table 1. These *P* values exceeded the significant level after Bonferroni correction with adjusted *P* value < 0.05, and their decimal place was changed to a single.

Supplementary Fig. 13 The significance of difference of 309 phenotypes between Hap1 and Hap2 of *MUC13*. Each point represents a phenotype. The y-axis denotes statistical significance tested by Student's test or Pearson's Chi-squared test. Numbers in brackets denote the number of traits. ETEC F4ac, Enterotoxigenic Escherichia coli F4ac; CSL, Carcass straight length; CDL, Carcass diagonal length; SIL, Small intestine length; FOCVL, Fourth cervical vertebra length; FICVL, Fifth cervical vertebra length; SICVL, Sixth cervical vertebra length; SECVL, Seventh cervical vertebra length; NBL, total length of cervical vertebra.

Fig. R1 QTL graphs for 10 traits presented in the Table 1 in the F2 population. (a) Detecting *MUC13* as a long-term balancing selection gene. GWASs for ETEC F4ac (b), carcass diagonal length (c), carcass straight length (d), anterior brachial bone length (e), total neck bone length (f), carcass weight (g), small intestine length (h), average daily gain at day 21 to 240 (i), left ear area (j), right ear area (k).

Table 1 The phenotypic difference between haplotypes of *MUC13*.

Trait	Hap1		Hap2		P Value	Additive effect	Dominance effect
	N	+/- Or Mean ± SD	N	+/- Or Mean ± SD			
ETEC F4ac adhesion	711	476(+)/235(-)	815	286(+)/529(-)	4.1×10^{-35} (OR = 3.75)	0.34	0.51
Carcass straight length (cm)	854	97.04 ± 6.87	1000	95.31 ± 7.52	2.3×10^{-07}	-1.85	-2.51
Carcass diagonal length (cm)	854	80.71 ± 5.89	1000	79.39 ± 6.41	3.9×10^{-06}	-1.41	-1.94
Small intestine length (m)	856	15.88 ± 1.93	1002	15.46 ± 2.33	3.2×10^{-05}	-0.44	-0.60
Fourth cervical vertebra length (cm)	854	2.03 ± 0.20	1002	1.98 ± 0.20	2.2×10^{-06}	-0.05	-0.06
Fifth cervical vertebra length (cm)	853	1.99 ± 0.19	997	1.95 ± 0.18	9.2×10^{-07}	-0.05	-0.06
Sixth cervical vertebra length (cm)	851	2.07 ± 0.20	995	2.03 ± 0.20	4.9×10^{-05}	-0.04	-0.06
Seventh cervical vertebra length (cm)	851	2.29 ± 0.21	999	2.23 ± 0.21	1.0×10^{-08}	-0.06	-0.08
Neck Bone Length (cm)	853	17.32 ± 1.41	1003	17.06 ± 1.49	9.1×10^{-05}	-0.28	-0.43

N, sample size; ETEC F4ac adhesion, susceptible to Enterotoxigenic *Escherichia coli* F4ac; +/- denotes susceptible/resistance to ETEC F4ac; OR, Odd ratio; Neck Bone Length, total length of cervical vertebra; Hap1 or Hap2, haplotypes of *MUC13*. The significance of ETEC F4ac phenotype difference between Hap1 and Hap2 was calculated by Pearson's Chi Squared test. The remaining phenotypic differences between Hap1 and Hap2 was calculated by Student's t test. The additive effect was estimated by recoding genotypes as 0, 1, 2. The dominance effect was estimated by recoding genotypes as 0, 1.

5. I did not find any statement where these haplotype data are made public which I assume is a prerequisite for publication.

Response: We have added it to the revised manuscript (main text: lines 756-759), and these haplotype data have now been released in National Genomics Data Center (NGDC, <https://ngdc.cnbc.ac.cn/gvm/>) with accession code: **GVM000479**.

Minor comments:

Line 27 and 180: I think it is more appropriate to refer to these as 1874 haploid genomes.

Response: We thank Reviewer #1 for this suggestion. These have been replaced (main

text: **lines 27, 141**) in the revised manuscript.

Line 79, Here you state that you sequenced representatives of most of the breeds (N=30) while on line 60 you state that there are possible more than 700 breeds.

Response: We thank Reviewer #1 for pointing out this inaccurate statement. It has been corrected to "we sequenced about 900 pig genomes covering most of the common breeds (N = 30) in Asia and parts of the worldwide breeds." in the revised manuscript (main text: **lines 81-82**).

Line 237-239. This text is essentially repeated on line 456-458. I suggest it is deleted here as it is better suited for the Discussion.

Response: We thank Reviewer #1 for pointing this out. We have deleted this in the section of *Results* and reworked it in the section of *Discussion* (main text: **lines 408-410**).

Line 471-472. The statement is unclear. Do you mean that it has been reported previously as a long-term balancing gene if so, add reference, or, if you mean that this is the first paper indicating long-term selection, please revise the text.

Response: We thank Reviewer #1 for pointing it out. We have corrected the statement to "To our knowledge, *MUC13* is firstly reported as a long-term balancing selection gene in our study." (main text: **lines 436-437**).

Discussion. There is some redundancy in the Discussion concerning the results on MUC13 and NPR3 which are discussed/summarized in three paragraphs of the Discussion and (in the Abstract). These overlaps should be eliminated.

Response: We appreciate Reviewer #1 for this suggestion. We have revised the section of *Discussion* concerning the results on MUC13 and NPR3 accordingly. We would like to Reviewer #1 again for reviewing the content about results on MUC13 and NPR3 in the section of *Discussion*.

Responses to Reviewer #2

Tong et al. used WGS data of 1096 pigs 43 breeds to develop a panel of haplotypes for imputation; carried out GWAS for intramuscular fat content and study a few gene in evolutionary contexts. Overall the manuscript is easy to follow. The standard methodologies commonly employed for such research were used.

Response: We thank Reviewer #2 for reviewing our manuscript and for this positive comment.

L108: After quality control, eight breeds had 1-2 samples. Is not this sample size too small to include a breed in the analysis?

Response: Among the 937 individuals mentioned in line 108 in the original manuscript, Creole pig, Gloucester old Spot, Iberian pig, Italian wild pig, Large Black, Leicoma, Mangalica, and Spanish wild boar had 1-2 samples (**Supplementary Table 1**). We agree with Reviewer #2 that using 1-2 samples to represent their breeds is insufficient in some analyses, such as population structure analysis and breed-based positive or balance selection analysis. But most of these breeds are in-danger aboriginal breeds and very difficult to sample. Following the reviewer's comment, we checked the influences on our results of these breeds: 1) In the population structure section, we excluded breeds with small sample size and used breeds with at least three samples for analysis. Thus, these eight breeds were not included in the population structure analysis. 2) In the balancing selection analysis, we scanned balancing selection loci across the following populations: Southeastern Chinese domestic pigs, Northwestern Chinese domestic pigs, Crossbred pigs, and European domestic pigs. Among these four populations, only European domestic pigs harbored six of these eight breeds (Creole pig, Gloucester old Spot, Iberian pig, Large Black, Leicoma, Mangalica). After excluding them, we again scanned balancing selection loci in European domestic pigs (**Fig. R2**), which was in line with the previous result (**Fig. 2a-c**). 3) In the positive selection analysis, two large-stature pigs (Duroc and Large white and Landrace and their crosses, Erhualian) and small-stature pigs (Bamaxiang, Tibetan) were used to detect candidate genes controlling body size, and these eight breeds were not included in them. The above results show that these sample size restricted eight breeds have not significantly affected the corresponding analyses. But the related data should be useful for genetic diversity and other genetics researches.

Fig. R3 Scanning the genome for balancing selection loci on autosomes in the European domestic pigs excluding Creole pig, Gloucester old Spot, Iberian pig, Large Black, Leicoma, and Mangalica (a). The nucleotide diversity (b) and Tajima's D (c) at MUC13 locus by a sliding 1 kb window.

Fig.2a. If I understand correctly, the plots are average of 10 imputed animals, only one round. The imputation should be repeated for large number of rounds including breeds which are not 'common'. Though it is important that the imputation accuracy is high for (common) commercial pigs, but generally, those have good reference populations. It is important to know the performance for breeds, which may not have good references.

Response: We thank Reviewer #2 for this constructive suggestion. We repeated the imputation for 470 rounds and estimated the accuracy for each breed as follows: 1) We sampled ten individuals as imputation targets and the remaining 1854 haplotypes as reference panel. The 50k, 60k, 80k, 100k, and 300k variants at autosomes were randomly selected to mimic chips. We then imputed the genotypes of unselected variants for sampled ten individuals. 2) Step 1) was repeated until genotypes of all individuals were imputed once. 3) The imputation accuracy was measured by average R^2 (squared Pearson correlation) between sequenced genotypes and imputed genotypes. The results show that LW, WDU, DU, LR, and PT have a high imputation accuracy in 41 breeds included in the reference panel (**Supplementary Fig. 8**). LWU, SUT, BMX, and EHL have a relatively high accuracy compared with Tibetan pigs (GST, YNT, SCT1, SCT2, TT). The imputation accuracy for wild pigs is relatively low, such as KRW, VIE, NTLW, SCW, ARSW, NCW, SPAW, ERSW, and ITAW. The above results have been added to the section of *Construction of the haplotype reference panel* (main text: **lines 158-163**) as follows: "Moreover, we estimated the imputation accuracy for each breed. The results showed that LW, WDU, DU, LR, and PT have a high imputation accuracy in 41 breeds included in the reference panel (Supplementary Fig. 8). LWU, SUT, BMX, and EHL have a relatively high accuracy compared with Tibetan pigs (GST, YNT, SCT1, SCT2, TT). The imputation accuracy for wild pigs is relatively low, such as KRW, VIE, NTLW, SCW, ARSW, NCW, SPAW, ERSW, and ITAW."

Supplementary Fig. 8 Genotypes imputation accuracy for each breed included in the haplotype reference panel. The letters on the upper left indicates the number of variants selected randomly for imputation. The y-axis represents imputation accuracy measured by R^2 (squared Pearson correlation). The x-axis denotes abbreviation of breeds. Each point represents an individual. ARSW, Asian Russia wild; ERSW, European Russia wild; BAM, Bamei; BAS, Baoshan; BMX, Bamaxiang; CRO, Creole; DU, Duroc; EHL, Erhualian; GOS, Gloucester old Spot; GST, Gansu Tibetan; HT, Hetao; IBR, Iberian pig; ITAW, Italian wild; JH, Jinhua; KRW, Korean wild; LA, Lean spotted pig; LB, Large Black; LCM, Leicoma; LR, Landrace; LUC, Luchuan; LW, LargeWhite; LWU, Laiwu; MGC, Mangalica; NCW, Northern chinese wild pig; NJ, Neijiang; NTLW, Netherland wild; PT, Pietrain; SCT, Sichuan Tibetan; SCW, Sourthern chinese wild; SPAW, Spanish wild; SUT, Sutai; TT, Tibetan Tibetan; VIE, Vietnam wild; WA, Wanan spotted; WDU, White Duroc; WZS, Wuzhishan; YCT, Yucatan; YNT, Yunnan Tibetan.

Fig.2b: Large increase in $-\log_{10}(p)$ with imputed WGS variants. Any explanation why some regions showed high increase in significance level while no changes in many others (ex. Chr2, 7, distal end of 9).

Response: The response for Reviewer #1 (please see **pages 2-4 of this response letter, specific comment 2**) can partly explain why five novel loci (on chr3, 5, 8, 9, and 13) showed high increase in significance level when using imputed WGS variants. Phenotypic value was significantly different (genome-wide threshold, $P < 5 \times 10^{-8}$) among haplotypes at each novel locus. These five novel loci emerged based on phenotypic variation across their haplotypes. Although these haplotypes harbor potential causative variants, the density of SNPs of 50k array data is relatively low, leading to a low Linkage Disequilibrium (LD) between SNPs and potential causative variants (**Supplementary Fig. 14**). Consequently, the statistical power of GWAS is reduced⁸, and significant loci could not be identified in chip GWAS. However, the number of variants increased dramatically after genotype imputation, which strengthens the LD between imputed WGS variants and potential causative variants (**Supplementary Fig. 14**), resulting in an increased statistical power of GWAS.

In the analysis of variance for haplotypes at the top locus (ex. Chr2, 7, distal end of 9) at each autosome in chip GWAS, the significances of 16 of 18 loci do not exceed the genome-wide threshold (**Table R1**), which suggested that these loci might not harbor potential causative variants for IMF. Therefore, their significance levels have little changes before and after imputation. The significance levels for loci on chromosomes 3 and 15 surpassed the genome-wide threshold in haplotype analysis. But the significance level of these two loci did not increase significantly in imputed GWAS. The haplotypes with the causative mutation are highly similar to that without the causative mutation in the population, which would result in IMPUTE5 software is hard to distinguish the correct haplotype to use when performing imputation⁹. Taken together, the high increase in significance level in imputation GWAS needs to satisfy two conditions as follows: 1) A locus harbors causative variants. 2) The number of variants increased after genotype imputation. No changes in significance level at other loci

might be the following reasons: 1) There is no causative variant at these loci. 2) True genotypes could not be imputed due to highly similar haplotypes or others.

The above discussions have been added to the section of *Discussion* (main text: **lines 401-407**) as follows: "The reason why significant loci could not be identified in the chip GWAS is that GWAS statistical power is reduced due to a low Linkage Disequilibrium (LD) between SNPs and potential causative variants (Supplementary Fig. 14)⁸. However, the number of variants increased dramatically after genotypes imputation, which strengthens the LD between imputed WGS variants and potential causative variants (Supplementary Fig. 14), resulting in an increased GWAS statistical power."

Supplementary Fig. 14 Average Linkage Disequilibrium between all variant pairs. The y-axis denotes the average LD degree measured by r^2 . The x-axis denotes the number of variants at chromosomes between two variants.

Fig.2c: Were the IMF phenotypes corrected for systematic fixed and random effects? The Xaxis level should be 'IMF phenotype'

Response: The IMF phenotypes were original data from Ding et al. study¹. This group released original phenotypic data and sex information, the *.fam file showed that all individuals were male. Therefore, we did not correct systematic fixed and random effects. Now, the "IMF" has been corrected to "IMF phenotype" in the Fig. 1c (previous Fig. 2c).

L248-250: "The correlation (R^2) between phenotypic value and imputation EBV ($R^2 = 0.67$) increased by 36.7% compared with chip EBV ($R^2 = 0.49$)". This is a remarkable improvement compared to several studies in livestock where sequence variants were included to increase genomic prediction accuracy (e.g. Xiang et al. 2021 Nat Commun., 10.1038/s41467-021-21001-0). Could you discuss the reasons to see such a high increase?

Response: Xiang et al. developed a bovine XT-50K genotyping array to increase accuracy in predicting genetic value of multiple important traits by integrating the functional, evolutionary and pleiotropic information of variants using GWAS, variant clustering and Bayesian mixture models¹⁰. The XT-50K custom array presented promising results in genomic prediction in independent datasets of 90,000+ dairy cattle. Averaged across three traits, the relative increase of the prediction accuracy for the XT-50K from the standard-50K ranged from 1.4% to 90% for four bovine breeds, with a median of 14%. The XT-50K array focuses on pleiotropic variants, which were pinpointed by multiple breeds, sexes, and traits. Therefore, it could be used for genomic prediction across multiple breeds, sexes, and traits. In our study, we think that there are several reasons why imputation EBV of IMF increased by 36.7% compared with chip EBV as follows: 1) Genotype imputation increases the number of variants in Linkage Disequilibrium with causal mutation or even the number of causal variants, resulting in identifying more novel QTLs. 2) Pre-selected variants by GWAS mitigate the effects of unrelated variants on genomic prediction. 3) These pre-selected variants were derived from GWAS of a single breed, sex, and trait. Except for pleiotropic variants, they probably harbor specific variants for the breed, sex, and trait. Although these variants may not suit other traits or breeds in genomic prediction, they have a good performance on the target breed and trait. 4) Furthermore, we conducted a 5-fold cross validation to evaluate the accuracy of genomic prediction. In brief, the 1490 samples are randomly partitioned into 5 equal sized subsamples. Of the 5 subsamples, a single subsample is retained for evaluating the accuracy, and the remaining 4 subsamples are used to perform GWAS for selecting variants at a $P = 0.01$ level. The cross-validation process is then repeated 5 times. The 5 results are averaged to produce a single estimation. The accuracy of imputation EBV ($R^2 = 0.45$) increased by 36.4% compared with chip EBV ($R^2 = 0.33$) (**Fig. R4**). However, the accuracy of imputation EBV from 5-fold cross validation decreased by 32.8% compared with that estimated by pre-selected variants from GWAS using all samples. It implied that the genomic prediction accuracy could be influenced by the sample size used for pre-selected variants. In the revised manuscript, we have added these discussions to the section of *Discussion* (main text: **lines 414-418, 426-435**).

Fig. R4 The accuracy of estimated breeding value (EBV) was measured by 5-fold cross validation. Significant variants at the $P < 0.01$ level from GWAS were used for estimating the breeding value of IMF by BLUP. The x-axis shows the estimated breeding value. The y-axis shows the phenotypic value of IMF. The accuracy of EBV is indicated by an R^2 (squared Pearson correlation) between phenotypic value and EBV.

L254: imputed EBV or selected imputed WGS variants?

Response: We have corrected "imputation EBV" to "selected imputed WGS variants" (main text: **lines 227-228**).

L266: ‘outstanding’ - > distinctive

Response: Has been corrected (main text: **line 240**).

L321: Validation of balancing selection gene MUC13 – I wonder if this was the right design to validate. “We found that only two (L334) haplotypes (Hap1 or Hap2) existed in this balancing selection block region in the F2 population”. Were there only 2 haplotypes in the parental population used to create F1 and what were the frequencies of haplotypes? What is the expectation of frequency changes in a ‘non-selected’ population in 2 generations?

Response: We thank Reviewer #2 for this very important comment. We summarized the frequencies of haplotypes within the balancing selection block in the F0, F1, and F2 populations genotyped (**Table R2**). There are two haplotype patterns with 38 haplotypes (No.Hap1=15, frequency=0.39; No.Hap2=23, frequency=0.61) in the parental population (F0). In the case of balancing selection, the frequency of Hap1 was expected to approach an intermediate level in the next generation. If the frequency of Hap1 is already at an intermediate level, it should be maintained. The frequency of Hap1 in the F1 generation is closer to intermediate level than in the F0 (**Table R2**). The frequency of Hap1 in the F2 is comparable to the F1. It implies that the frequency of

Hap1 was driven to a relatively intermediate level from F0 to F1 and maintained from F1 to F2. However, the number of generations (only two generations) being analyzed is too less to support our conclusion absolutely from statistics. We have deleted sentences "Same as in balancing selection scanning populations, we found that only two haplotypes (Hap1 or Hap2) existed in this balancing selection block region in the F2 population (Fig. 3e). The frequency of older Hap1 also stays at an intermediate level (0.46)." in the revised manuscript. There is no doubt that this revision is very important, but it has little influence on the identification of balancing selection genes.

Table R2 The frequencies of Hap1 and Hap2 in the F0, F1, and F2 populations.

Population	No.individual	Freq.Hap1	Freq.Hap2
F0	19	0.39	0.61
F1	44	0.47	0.53
F2	1020	0.46	0.54

No.individual, the number of individuals; Freq.Hap1, the frequency of Hap1; Freq.Hap2, the frequency of Hap2.

Table 1. ETEC F4ac – Please specify ‘+’ and ‘-’. what test did you do (McNemar’s test)? Odd ratio may be informative.

Response: +/- denotes susceptible/resistance to ETEC F4ac. We have added this illustration in the Table 1. The test for ETEC F4ac was Pearson's Chi Squared test, which has been illustrated in the explanation of Table 1. We also calculated the Odd ratio for ETEC F4ac phenotype and haplotypes of *MUC13*, and it have been added to Table 1.

Fig.4e: I do not understand this figure; more details required.

Response: We are sorry for that and thank Reviewer #2 for this comment. After considering the comment of Reviewer #1 (please see **pages: 5-9 of this response letter**), we have deleted Fig. 4e in the revised manuscript.

L556: Minor allele count ≤ 2 ; is it not too liberal?

Response: We thank Reviewer #2 for this comment. The outgroups and *Sus scrofa* were used to call genotypes jointly. They have many outgroup-specific (or species-specific) variants compared with *Sus scrofa*. These variants provide less reference to *Sus scrofa*. Because the outgroups are diploid. We thus used criteria of Minor allele count ≤ 2 to exclude these outgroup-specific variants to reduce data volume.

Responses to Reviewer #3

In their manuscript Tong et al. describe the development of a genetic resource for pigs, that will provide higher power and resolution in GWAS and genomic Selection. The strength of the new resource is shown for the identification of potential genes involved in quantitative traits (e.g. IMF) and/or under selection (MUC13 and NPR3). The manuscript in general is written clearly, although there are some sections where the English language can be improved (see minor comments for details).

Response: We thank Reviewer #3 very much for the thorough evaluation of our work and kindly positive comments.

The literature cited seems to be somewhat biased and it gives the impression that this might have been done to strengthen the “innovativeness of the manuscript”. E.g. in the introduction (lines 74-75) it is stated that it is not known whether pig-specific balancing selection genes exist, while a publication in PLoS genetics clearly showed balancing selection for a 200 Kb deletion affecting the BMPER and PPS9 genes in pigs (Derks et al. PLoS gen. 2018, 14: e1007661). Another example is a reference to unpublished results of the authors citing different evolutionary time points than what is generally accepted and supported by many previous other studies. The authors use 219 Kya for the last common ancestor of European and Asian wild boar, while previous studies estimate this to be ~ 1 Mya. A similar discrepancy is seen for the divergence of the Sus species which the authors claim to be 1.36 Mya while multiple other studies estimate this to be ~3.5-4 Mya. The latter estimates are not only derived from genetic studies but are also supported by fossil records.

The same for the results and discussion about the improved resolution and power for GWAS when using imputed SNPs. This has been shown in previous studies (e.g. a recent study for pigs is described by Derks et al 2021, Genomics 113: 2229).

Response: We have paid particular attention to this aspect in our manuscript and read the studies mentioned in the comment. Our study mainly focuses on the long-term balancing selection genes whose selection pressures are mainly from the natural environment. We revised our statement in lines 74-75 in the original manuscript from the "Whether pig-specific balancing selection genes exist or not remains elusive." to "Although several balancing selection genes have been discovered in studies of pigs^{11, 12}, they were found in breeding populations selected by favorable traits in a short term. For example, balancing selection was identified for a 200 kb deletion affecting the *BMPER* and *PPS9* genes a breeding population of pigs¹¹. However, whether pig-specific long-term balancing selection genes exist or not remains elusive." in the revised manuscript (main text: **lines 73-78**).

Regarding the evolution time of pigs, we have published the relevant research results based on our specific designed and tested de novo mutation rate of 3.6×10^{-9} per generation in *Genomics Proteomics Bioinformatics* in 2022 (doi: 10.1016/j.gpb.2022.02.001). We estimated de novo mutation rate of pigs using whole-genome sequencing data from nine individuals in a three-generation pedigree through

highly stringent filtering and validation. Using this mutation rate, we re-investigated the evolutionary history of pigs. Our results suggested that *Sus* speciation occurred ~ 1.36 million years ago (Mya); European pigs split from Asian pigs only ~ 219 Kya; South and north Chinese wild pigs split ~ 25 Kya. The evolutionary time of pigs estimated based on novel de novo mutation rate of pigs was different with and more recent than the previously generally accepted evolutionary time of pigs that was not only derived from genetic studies but are also supported by fossil records¹³. Therefore, the evolutionary time of pigs is still controversial, but these viewpoints about pig evolutionary time are not the focus of this study. Therefore, we have deleted the statements about concrete divergence times and reworked the statements (main text: **lines 271-273, 454-457**).

We have carefully read the study described by Derks et al.¹⁴. Using 552,000 imputed SNPs, the GWAS analysis revealed 271 QTL regions for 83 traits, showing great ability in improving resolution and power for GWAS. Moreover, Derks et al. prioritize variation by predicted variant impact scores (pCADD), functional genomic information, and associated phenotypes in other mammalian species. Thus, in the results and discussion about the improved resolution and power for GWAS, we have reworked the statements as follows: 1) The sentence "To assess GWAS power and detection ability by including the haplotype reference panel in an unrelated population with low-density SNP chips." was changed to "Previous studies show that the resolution and power for GWAS could be dramatically improved by including haplotype reference panel^{14, 15}. To assess the ability of GWAS power and detection ability by including our haplotype reference panel in an unrelated population with low-density SNP chips." (main text: **lines 189-192**). 2) The sentence "The above results suggested the huge application of the haplotype reference panel for reanalyzing previously published array data, such as for GWAS and fine mapping." was changed to "The above results suggested that our haplotype reference panel has huge application in reanalyzing previously published array data." (main text: **lines 213-215**). 3) The sentence "By reanalyzing a previously published chip-based GWAS study of IMF, our haplotype reference panel greatly boosted the GWAS power and the accuracy of EBV, indicating the potential use of our haplotype panel in the rediscovery of the previously genotyped low-density SNP array data." was changed to "By reanalyzing a previously published chip-based GWAS study of IMF, our haplotype reference panel greatly boosted the GWAS power and the accuracy of EBV, indicating the potential use of haplotype panels in the rediscovery of the previously genotyped low-density SNP array data, in line with the previous study¹⁴." (main text: **lines 372-376**).

Line 350-351: "but benefits growth-related traits, which supports the hypothesis of balancing selection." I don't think this statement is correct. This would be the case if this is observed in a population that is under selection for high growth like is the case for domestic breeds" However, this is not the case for wild boar populations. Why would the Hap2 haplotype be under selection in all wild *Sus scrofa* populations (and not in the other suids, many of which even are similar in size as wild boar). The same applies for the statement made in lines 385-386.

Response: We agree with the reviewer that high growth does not lead to the selection of wild boar. *MUC13* was highly expressed in the small intestine of pigs¹⁶. The small intestine at which ETEC F4ac interacts exerts an effect on nutrient absorption¹⁷. We correlated haplotypes of *MUC13* with 309 phenotypes to explore potential traits under selection (please see **pages: 5-9 of this response letter**). The *MUC13* is significantly associated with growth-related traits. Thus, we speculated that high growth probably is the consequence of good nutrient absorption. In an environment where food is relatively scarce, obtaining more nutrients will increase the survival chances of wild boar and provide the basis for sexual maturation. The good nutrient absorption from foods in the natural environment would increase the probability of that wild boar passing its DNA to the next generation. Therefore, the nutrient absorption capacity may lead to the selection of wild boar and bring high growth incidentally. Considering this insightful suggestion, we have corrected lines 350-351 to "Collectively, the Hap1 confers diminished resistance to ETEC F4ac but is positively associated with growth-related traits." and added the above discussions to the section of *Discussion* (main text: **lines 319-320**).

The large stature pigs (pDLY: Duroc and Large white and Landrace and their crosses, EHL: Erhualian) and small stature pigs (BMX: Bamaxiang, TBT: Gansu and Sichuan and Tibetan and Yunnan Tibetan) mentioned in lines 385-386 of the original manuscript are domestic pigs¹⁸, which have been selected for some favorable traits, such as large stature. A previous study showed that high EHHs of haplotypes could provide evidence of positive selection on this locus¹⁹. In our results, pDLY and EHL have a higher EHHs than in BMX and TBT. Thus, we thought that the *NPR3* locus had undergone a positive selection in large stature pigs (pDLY, EHL). The statement in lines 385-386 has been corrected to "High EHHs of haplotypes could provide evidence of positive selection on this locus¹⁹. Remarkably, the EHHs were higher in pDLY and EHL than BMX and TBT (Fig. 3c), indicating positive selection on the locus in large stature pigs." in the revised manuscript, and the reference about EHH principle has been added to aid the statement (main text: **lines 352-355**).

Lines 484-486: I do not agree with the statement that the observed frequency of 0.58 is consistent with balancing selection. The allele frequencies in the parental populations are already around 50% and the authors state that there was no selection applied in the F1 and the F2 generations. So an observed allele frequency of Hap1 of 0.58 in the F2 population is what you would expect.

Response: We thank Reviewer #3 for this helpful comment. There may be an unclear statement with lines 484-486. Now, we have corrected the statement "We found only two haplotypes (Hap1 or Hap2) in our 941 analytic individuals, and the frequency of Hap1 stays at an intermediate level (0.58), again in consist with the characteristics of balancing selection." to "We found only two haplotypes (Hap1 or Hap2) in the populations used for scanning long-term balancing selection genes (SCD, NCD, Cross, EUD). The frequency of Hap1 stays at an intermediate level (0.58), again in consist

with the characteristics of balancing selection." If I understand correctly, the reviewer means that an observed allele frequency of Hap1 of 0.46 in the F2 population (lines 333-336) could not provide evidence for balancing selection of *MUC13* because the allele frequencies in the parental populations are already around 50%. Reviewer #2 also raised a very similar comment (please see **pages: 17-18 of this response letter**). To avoid repetitive responses, please see it for details. Finally, we deleted sentences "Same as in balancing selection scanning populations, we found that only two haplotypes (Hap1 or Hap2) existed in this balancing selection block region in the F2 population (Fig. 3e). The frequency of older Hap1 also stays at an intermediate level (0.46)." in the revised manuscript. There is no doubt that this revision is very important, but it has little influence on the identification of balancing selection genes.

Line 552: Removing only SNPs that are within 1 bp of an INDEL seems rather relaxed and still potentially leads to inclusion of false positives.

Response: We thank Reviewer #3 for this helpful comment. We used 325 individual array genotypes to evaluate the accuracy of SNPs that are within 1-50 bp of Indels (**Supplementary Fig. 15**). We found that the accuracy of SNPs reaches saturation at a distance of 5bp from Indels, with an average of 98.2%. Therefore, we removed SNPs that are within 4 bp of Indels in the genotypic files and have corrected the filtering criteria in line 517 of the revised main text. The data of haplotype reference panel had been re-uploaded to National Genomics Data Center (NGDC, <https://ngdc.cncb.ac.cn/gvm/>) with accession code: **GVM000479**. This result has been added to the section of *Methods* (main text: **lines 514-517**) as follows: "Due to the low quality of SNP near INDEL, we used 325 individual array genotypes to evaluate the accuracy of SNPs that are within 1-50 bp of Indels (Supplementary Fig. 15). The accuracy of SNPs reaches saturation at a distance of 5bp from Indels, we thus removed the SNP within 4 bp of INDEL using bcftools v1.9.".

Supplementary Fig. 15 The accuracy of SNPs that are within 1-50 bp of INDELS. Each triangle in red represents a set of SNPs near Indels. The y-axis shows the concordance rate between sequence genotypes and array genotypes.

Minor comments:

Lines 78-79: The authors state that the individuals used in their study cover most pig breeds in Asia and parts of the breeds worldwide. This contradicts an earlier statement made in lines 60-61 that there are 730 pig breeds worldwide of which two-third are found in China and Europe.

Response: We thank Reviewer #3 for pointing out this. It has been corrected to "we sequenced about 900 pig genomes covering most of the common breeds (N = 30) in Asia and parts of the worldwide breeds." (main text: **lines 81-82**).

Line 78: insert "we" before "sequenced"

Response: Has been corrected (main text: **line 81**).

Line 78: Change "covered" by "covering"

Response: Has been corrected (main text: **line 81**).

Line 115: delete "is" in "rate is ranged"

Response: Has been deleted (main text: **line 122**).

Lines 117 and 119: "Mendel error" should be "Mendelian error"

Response: Have been corrected (main text: **lines 124-125**).

In lines 152-164: Remove "the" before the abbreviations SCT1 and LA. Als change "was" clustered to "clustered. So instead of (line 155) "The STC1 was clustered" write "STC1 clustered".

Response: We thank Reviewer #3 for this comment. Following the suggestions of Reviewer #1 (please see **page 2 of this response letter**), we have deleted this section of *Population structure* and the related discussion in the revised main text. Thus, the statement in lines 152-164 was also deleted.

I although think it sounds better if the first time an abbreviation is used to first write it in full: E.g. in line 152-153, change "Although the SCT1 (Luding Tibetan pigs, N = 50) and the SCT2 (Litang Tibetan pigs, N = 12) lived in" I suggest to write "Although Luding Tibetan pigs (SCT1, N = 50) and Litang Tibetan pigs (SCT2, N = 12) lived in"

Response: We thank Reviewer #3 for this comment. Following the suggestions of Reviewer #1 (please see **page 2 of this response letter**), we have deleted this section of *Population structure* and the related discussion in the revised main text. Thus, the statement in lines 152-153 was also deleted.

Line 163: Change “The GST (Gansu Tibetan pigs, N = 14) was located” to “Gansu Tibetan pigs (GST, N = 14) are located”

Response: We thank Reviewer #3 for this comment. Following the suggestions of Reviewer #1 (please see **page 2 of this response letter**), we have deleted this section of *Population structure* and the related discussion in the revised main text. Thus, the statement in lines 163 was also deleted.

Line 171: insert “a” before “haplotype”

Response: Has been corrected (main text: **line 132**).

Line 174: Change “built” to “build”

Response: Has been corrected (main text: **line 135**).

Line 187” Change “As” to “Because of”

Response: Has been corrected (main text: **line 148**).

Line 198: “elected” should be “selected”

Response: Has been corrected (main text: **line 165**).

Lines 231-232: Insert “the” before “chip”

Response: Has been corrected (main text: **line 201**).

Line 232: Change “Of which three leading” to “Three leading.....”

Response: Has been corrected (main text: **line 202**).

Line 232: Change “gene” to “genes”

Response: Has been corrected (main text: **line 202**).

Line 274: Change “Expectedly” to “As expected”

Response: Has been corrected (main text: **line 248**).

Line 335: Insert “the” before “older”

Response: We thank Reviewer #3 for this comment. Following the suggestions of

Reviewer #2 (please see **pages 18 of this response letter**), the statement in line 335 has been deleted.

Line 344: “Compared” instead of “compare”

Response: We thank Reviewer #3 for this comment. The statement in line 344 has been reworked (main text: **lines 309-312**).

Line 364: Change “varied” to “varies”

Response: Has been corrected (main text: **line 332**).

Lines 425-428: There are multiple errors in this sentence. Change this sentence to “Further analyses in various Tibetan pig populations show that the Luding Tibetan pigs (STC1) and the Litang Tibetan pigs (STC2) did not cluster together in the Neighbor-Joining tree and have distinct ancestral compositions, although they live in the same geographical location.”

Response: We thank Reviewer #3 for this comment. Following the suggestions of Reviewer #1 (please see **page 2 of this response letter**), we have deleted this section of *Population structure* and the related discussion in the revised main text. Thus, the statements in lines 425-428 were also deleted.

Lines 429-430: Again remove “the” before STC1.

Response: We thank Reviewer #3 for this comment. Following the suggestions of Reviewer #1 (please see **page 2 of this response letter**), we have deleted this section of *Population structure* and the related discussion in the revised main text. Thus, the statement in lines 429-430 was also deleted.

Line 429: Change “artificial” to “artificially”

Response: We thank Reviewer #3 for this comment. Following the suggestions of Reviewer #1 (please see **page 2 of this response letter**), we have deleted this section of *Population structure* and the related discussion in the revised main text. Thus, the statement in line 429 was also deleted.

Line 449: Replace “their” by “the”

Response: Has been corrected (main text: **line 398**).

Line 453: Insert “the” before “Porcine”

Response: We thank Reviewer #3 for this comment. Following the suggestions of

Reviewer #2 (please see **pages 14-15 of this response letter**), the statement in line 453 has been reworked.

Lines 457-458: I suggest connecting these two sentences. “.....Black cattle and GDF-3 is”

Response: Has been corrected (main text: **lines 408-410**).

Line 485: Delete “analytic”

Response: Has been deleted (main text: **lines 449-451**).

Line 510: Change “under a positive selection” to “is under positive selection”

Response: We thank Reviewer #3 for this comment. Reviewer #1 raised a comment (please see **page 10 of this response letter**) that there is some redundancy in the Discussion concerning the results on MUC13 and NPR3. Thus, we have deleted statement in line 510 in the revised manuscript.

Line 510: Insert “the” before “major”

Response: Has been corrected (main text: **line 478**).

Reference

1. Ding, R., *et al.* Single-locus and multi-locus genome-wide association studies for intramuscular fat in Duroc pigs. *Front. Genet.* **10**, 619 (2019).
2. Chen, F.-C. & Li, W.-H. Genomic divergences between humans and other hominoids and the effective population size of the common ancestor of humans and chimpanzees. *The American Journal of Human Genetics* **68**, 444-456 (2001).
3. Liu, H., *et al.* A Single-Step Genome Wide Association Study on Body Size Traits Using Imputation-Based Whole-Genome Sequence Data in Yorkshire Pigs. *Front. Genet.* **12**, 629049 (2021).
4. Estrada, K., *et al.* A genome-wide association study of northwestern Europeans involves the C-type natriuretic peptide signaling pathway in the etiology of human height variation. *Hum. Mol. Genet.* **18**, 3516-3524 (2009).
5. Soranzo, N., *et al.* Meta-analysis of genome-wide scans for human adult stature identifies novel Loci and associations with measures of skeletal frame size. *PLoS Genet.* **5**, e1000445 (2009).
6. Jaubert, J., *et al.* Three new allelic mouse mutations that cause skeletal overgrowth involve the natriuretic peptide receptor C gene (*Npr3*). *Proc. Natl. Acad. Sci. U.S.A.* **96**, 10278-10283 (1999).
7. Takasuga, A. *PLAG1* and *NCAPG-LCORL* in livestock. *Anim. Sci. J.* **87**, 159-167 (2016).
8. Wang, W., Barratt, B. J., Clayton, D. G. & Todd, J. A. Genome-wide association studies: theoretical and practical concerns. *Nature Reviews Genetics* **6**, 109-118 (2005).
9. Yan, G., *et al.* An imputed whole-genome sequence-based GWAS approach pinpoints causal mutations for complex traits in a specific swine population. *Sci. China Life Sci.* **65**, 781-794 (2022).
10. Xiang, R., *et al.* Genome-wide fine-mapping identifies pleiotropic and functional variants that predict many traits across global cattle populations. *Nat. Commun.* **12**, 1-13 (2021).
11. Derks, M. F., *et al.* Balancing selection on a recessive lethal deletion with pleiotropic effects on two neighboring genes in the porcine genome. *PLoS Genet.* **14**, e1007661 (2018).
12. Matika, O., *et al.* Balancing selection at a premature stop mutation in the myostatin gene underlies a recessive leg weakness syndrome in pigs. *PLoS Genet.* **15**, e1007759 (2019).
13. Zhang, M., Yang, Q., Ai, H. & Huang, L. Revisiting the evolutionary history of pigs via De Novo mutation rate estimation in a three-generation pedigree. *Genomics Proteomics Bioinformatics*, <https://doi.org/10.1016/j.gpb.2022.1002.1001> (2022).
14. Derks, M. F., *et al.* Accelerated discovery of functional genomic variation in pigs. *Genomics* **113**, 2229-2239 (2021).
15. Yan, G., *et al.* An imputed whole-genome sequence-based GWAS approach pinpoints causal mutations for complex traits in a specific swine population. *Sci. China Life Sci.* **65**, 781-794 (2021).
16. Zhang, B., *et al.* Investigation of the porcine *MUC13* gene: isolation, expression, polymorphisms and strong association with susceptibility to enterotoxigenic *Escherichia coli* F4ab/ac. *Anim. Genet.* **39**, 258-266 (2008).
17. Tharakan, A., Norton, I., Fryer, P. & Bakalis, S. Mass transfer and nutrient absorption in a simulated model of small intestine. *J. Food Sci.* **75**, E339-E346 (2010).
18. Wang, L., *et al.* Animal genetic resources in China: pigs. *China Agric Ture Press* **5**, 25-29 (2011).

19. Sabeti, P. C., *et al.* Detecting recent positive selection in the human genome from haplotype structure. *Nature* **419**, 832-837 (2002).

REVIEWER COMMENTS

Reviewer #1 (Remarks to the Author):

The authors have addressed the comments I made on the first version. I have only two further comments on data presentations that should be clarified.

Figure 2e. This figure is supposed to illustrate the presence of only two haplotype blocks among these individuals. This needs to be explained better, and explain what the rows and columns stands for in this figure. I assume these are haplotypes sorted by type. An alternative way would be to sort them by population and use genotypes instead of haplotypes, then if there is complete LD all SNPs will be either all homozygous alleles associated with haplotype 1 or 2 or all SNPs will be heterozygous at all positions. A third option would be to report all haplotype frequencies in a table across breeds to show that the frequencies are remarkably stable.

Table 1. I don't fully understand the QTL analysis here. It is not clear to me why you estimate the haplotype effect rather than just the additive effects estimated as the difference between the two homozygotes/2. Furthermore, for all traits you report complete dominance or overdominance, i.e., the estimated d is equal or higher than a . You should explain polarity how you estimated a and d . Does this mean that the incidence of ETEC is highest or lowest in the heterozygotes? I think this is worth discussing in the paper. I think this is of interest if this polymorphism is maintained as a balanced polymorphisms because overdominance is a classical mechanism causing balanced polymorphisms.

Reviewer #2 (Remarks to the Author):

I will like to thanks authors for extensive revision and I am satisficed with the revised manuscript.

Reviewer #3 (Remarks to the Author):

I thank the authors for the very thorough rebuttal and changes made to address the comments of the reviewers. I have only a few minor comments that need to be addressed.

Line 78: Change PPS9 gene to BBS9 gene (my apologies, I used the incorrect name myself in my review of the original version of the manuscript)

In the abstract lines 27-28 it is stated that: “we accurately constructed a panel of 1,874 haplotypes haploid 28 genomes with 41,964,356 genetic variants. However, in the results (lines 117-120) a larger number of genetic variants is mentioned: “Finally, 937 individuals ($N \geq 1$, No.breeds = 41; $N \geq 3$, No.breeds = 33) and 45,191,157 variants (41,326,535 SNPs; 3,864,622 Indels) were used for further analyses. Then in lines 181-183 it again is mentioned that: “The 937 high-quality individuals and 41,964,356 autosomal variants (38,483,119 SNPs; 3,481,237 Indels) were used for constructing a haplotype reference panel.”

Line 329: “suggests that Hap2 may originate from an extinct species (*Sus scrofa*)”. “(*Sus scrofa*) should be deleted as this is not the suggested extinct species, but the recipient species.

Furthermore the alternative suggestion (line 330-331) “or occur at a too low frequency in other species to be detected by the small sample sizes used” seems highly unlikely.

Line 580-581: “again in consist with the characteristics of balancing selection”. This sentence is not correct change to either “again consistent with” or “again in agreement with”.

Response to Reviewers

March 9, 2023

We carefully checked the comments and revised the paper by point to point. All revisions were highlighted in red in the manuscript with track. The point-by-point responses to the concerns are listed as follows.

Responses to Reviewer #1

The authors have addressed the comments I made on the first version. I have only two further comments on data presentations that should be clarified.

Response: We greatly thank Reviewer #1 for his/her precious time in reviewing our paper.

Figure 2e. This figure is supposed to illustrate the presence of only two haplotype blocks among these individuals. This needs to be explained better, and explain what the rows and columns stands for in this figure. I assume these are haplotypes sorted by type. An alternative way would be to sort them by population and use genotypes instead of haplotypes, then if there is complete LD all SNPs will be either all homozygous alleles associated with haplotype 1 or 2 or all SNPs will be heterozygous at all positions. A third option would be to report all haplotype frequencies in a table across breeds to show that the frequencies are remarkably stable.

Response: We thank Reviewer #1 for this comment. Figure 2e was used to show that only two haplotype patterns exist among these individuals. To present it more clearly, we have reworked **Figure 2e** and corrected the legend to "Two haplotype patterns inside the haplotype block of *MUC13* in populations EUD, NCD, SCD, and Cross. Rows represent haplotypes sorted by type and population. Columns denote variants from position chr13:135413350 to 135421391. The lattices in red/green indicate reference/alternative alleles." (main text: **lines 286-289**). We took the reviewer's third option to report haplotype frequencies across breeds harboring at least three individuals (**Supplementary Table 5**). We found that the frequencies of Hap1 in most breeds stay at an intermediate level (0.3~0.7). NJ, LWU and HT have a high frequency of Hap1 (0.92, 0.97 and 1.00, respectively). We have added the sentence "The frequencies of Hap1 are relatively stable across breeds (Supplementary Table 5)." to the result section (main text: **lines 258-259**).

Figure 2e Two haplotype patterns inside the haplotype block of *MUC13* in populations EUD, NCD, SCD, and Cross. Rows represent haplotypes sorted by type

and population. Columns denote variants from position chr13:135413350 to 135421391. The lattices in red/green indicate reference/alternative alleles.

Supplementary Table 5 The frequency of Hap1 of *MUC13* across breeds harboring at least three individuals in populations EUD, NCD, SCD, and Cross. EUD, European domestic pigs; NCD, Northwestern Chinese domestic pigs; SCD, Southeastern Chinese domestic pigs; Cross, European and Eurasian crossbred pigs.

Population	Breed	Sample size	Frequency of Hap1	
EUD	DU	29	0.52	0.67
EUD	PT	6	0.67	
EUD	LW1	62	0.69	
EUD	LW2	5	0.70	
EUD	LR	25	0.74	
EUD	YCT	11	0.86	
NCD	MIN	6	0.33	0.77
NCD	SCT1	50	0.56	
NCD	BAM	6	0.58	
NCD	TT	12	0.67	
NCD	LWU	75	0.97	
NCD	HT	6	1.00	
SCD	LUC	6	0.33	0.44
SCD	BMX	84	0.39	
SCD	EHL	132	0.40	
SCD	SCT2	12	0.42	
SCD	GST	14	0.50	
SCD	JH	6	0.50	
SCD	WZS	6	0.50	
SCD	WA	6	0.67	
SCD	YNT	12	0.79	
SCD	NJ	6	0.92	
Cross	F2	52	0.46	0.51
Cross	F1	44	0.47	
Cross	F3	44	0.47	
Cross	SUT	63	0.48	
Cross	DLY	43	0.52	
Cross	WDU	10	0.65	
Cross	LA	9	0.83	

BAM, Bamei; BAS, Baoshan; BMX, Bamaxiang; DU, Duroc; EHL, Erhualian; F1/F2/F3, Eurasian Crossbred; GST, Gansu Tibetan; HT, Hetao; JH, Jinhua; LA, Lean spotted pig; LR, Landrace; LUC, Luchuan; LW, LargeWhite; LWU, Laiwu; NJ, Neijiang; PT, Pietrain; SCT, Sichuan Tibetan; SUT, Suta; TT, Tibetan Tibetan; WA, Wanan spotted; WDU, White Duroc; WZS, Wuzhishan; YCT, Yucatan; YNT, Yunnan Tibetan.

Table 1. I don't fully understand the QTL analysis here. It is not clear to me why you estimate the haplotype effect rather than just the additive effects estimated as the difference between the two homozygotes/2. Furthermore, for all traits you report complete dominance or overdominance, i.e., the estimated d is equal or higher than a . You should explain polarity how you estimated a and d . Does this mean that the incidence of ETEC is highest or lowest in the heterozygotes? I think this is worth discussing in the paper. I think this is of interest if this polymorphism is maintained as a balanced polymorphisms because overdominance is a classical mechanism causing balanced polymorphisms.

Response: We thank Reviewer #1 for this helpful comment. We identified an 8kb haplotype block under balancing selection. But the causative variant remains unknown. Thus, we estimated the haplotype effect to represent the effect of unknown causative variant. A complete additive regression model was used to estimate the additive effect as follows:

$$Y = B + A * W1$$

Y denotes the phenotypic value. B denotes intercept. $W1$ represents the genotypes of haplotypes, with values 0, 1, and 2 for, respectively, Hap1/Hap1, Hap1/Hap2, and Hap2/Hap2. A represents the estimated additive effect.

Analogously, a complete dominance regression model was used to estimate the dominance effect as follows:

$$Y = B + D * W2$$

$W2$ denotes the genotypes of haplotypes, with values 0, 0, and 1 for, respectively, Hap1/Hap1, Hap1/Hap2, and Hap2/Hap2. D represents the estimated dominance effect. Following the reviewer's suggestion and referring to two studies of dominance effect^{1, 2}, we selected a proxy SNP 13_135417839, in complete linkage disequilibrium with the haplotype of *MUC13*, to estimate the additive and dominance effects of this locus using the following model:

$$Y = B + A * X1 + D * X2$$

$X1$ denotes the genotypes (A/A, A/G, G/G) of SNP 13_135417839. A/A, A/G, and G/G were recoded as 2, 1, and 0. A represents the estimated additive effect. To account for dominance, construct another new variable, $X2$, with values of 0, 1, and 0 for, respectively, genotypes A/A, A/G, and G/G. D represents the estimated dominance effect. The additive and dominance effects of this SNP have been added to **Table 1**, and the corresponding footnotes have been reworked (main text: **lines 328-337**).

Besides, we calculated the incidence of ETEC in the above three genotypes of SNP 13_135417839. The incidence of ETEC for genotypes AA, AG, and GG are 0.78, 0.58, and 0.13, respectively. Therefore, the ETEC is not overdominance. We have added

corresponding discussion to the section Discussion (main text: **lines 469-480**) as follows: "There are two common mechanisms that cause balancing selection: 1) Overdominance occurs, controlled by one trait, where the fitness of the heterozygote is superior to either of the homozygotes. 2) Pleiotropy, multiple traits are governed by a single locus, gives the heterozygote the highest fitness. To inspect whether overdominance occurred in ETEC F4ac, we selected a proxy SNP 13_135417839, in complete linkage disequilibrium with the haplotype of *MUC13*, to estimate the overdominance of ETEC F4ac. The incidence of ETEC F4ac for genotypes AA, AG, and GG of SNP 13_135417839 are 0.78, 0.58, and 0.13, respectively, indicating that it is not the case of overdominance. Thus, we speculated the *MUC13* pleiotropy might cause this balancing selection. Our further analyses indeed show that *MUC13* associates with ETEC F4ac and 8 growth-related traits, confirming its pleiotropy. We then investigate how *MUC13* might affect the growth-related traits."

Table 1 The phenotypic difference between haplotypes of *MUC13*.

Trait	Hap1		Hap2		P Value	Additive effect	Dominance effect
	N	+/- Or Mean ± SD	N	+/- Or Mean ± SD			
ETEC F4ac adhesion	711	476(+)/235(-)	815	286(+)/529(-)	4.1×10^{-35} (OR = 3.75)	-0.328**	-0.130**
Carcass straight length (cm)	854	97.04 ± 6.87	1000	95.31 ± 7.52	2.3×10^{-07}	1.823**	0.255
Carcass diagonal length (cm)	854	80.71 ± 5.89	1000	79.39 ± 6.41	3.9×10^{-06}	1.384**	0.249
Small intestine length (m)	856	15.88 ± 1.93	1002	15.46 ± 2.33	3.2×10^{-05}	0.429**	0.075
Fourth cervical vertebra length (cm)	854	2.03 ± 0.20	1002	1.98 ± 0.20	2.2×10^{-06}	0.048**	0.001
Fifth cervical vertebra length (cm)	853	1.99 ± 0.19	997	1.95 ± 0.18	9.2×10^{-07}	0.045**	0.007
Sixth cervical vertebra length (cm)	851	2.07 ± 0.20	995	2.03 ± 0.20	4.9×10^{-05}	0.040**	0.006
Seventh cervical vertebra length (cm)	851	2.29 ± 0.21	999	2.23 ± 0.21	1.0×10^{-08}	0.059**	0.004
Neck Bone Length (cm)	853	17.32 ± 1.41	1003	17.06 ± 1.49	9.1×10^{-05}	0.272**	0.109

N, sample size; ETEC F4ac adhesion, susceptible to Enterotoxigenic Escherichia coli F4ac; +/- denotes susceptible/resistance to ETEC F4ac; OR, Odd ratio; Neck Bone Length, total length of cervical vertebra; Hap1 or Hap2, haplotypes of *MUC13*. The significance (*P* Value) of ETEC F4ac phenotypic difference between Hap1 and Hap2 was calculated by Pearson's Chi Squared test. The significances (*P* Value) of remaining phenotypic differences between Hap1 and Hap2 was calculated by Student's *t* test. Proxy SNP 13_135417839, in complete linkage disequilibrium with the haplotype of *MUC13*, was used to estimate the additive and dominance effects with model $Y = B + A * X1 + D * X2$. *Y* denotes the phenotypic value. *B* denotes intercept. *X1* represents the

genotypes (A/A, A/G, G/G) of SNP 13_135417839. A/A, A/G, and G/G were recoded as 2, 1, and 0. A represents the estimated additive effect. To account for dominance, construct another new variable, X2, with values of 0, 1, and 0 for, respectively, genotypes A/A, A/G, and G/G. D represents the estimated dominance effect. ** represents that p-value is less than 0.01.

Responses to Reviewer #2

I will like to thanks authors for extensive revision and I am satisfied with the revised manuscript.

Response: We thank Reviewer #2 for his/her valuable suggestions.

Responses to Reviewer #3

I thank the authors for the very thorough rebuttal and changes made to address the comments of the reviewers. I have only a few minor comments that need to be addressed.

Response: We thank Reviewer #3 very much for the evaluation of our revised manuscript.

Line 78: Change PPS9 gene to BBS9 gene (my apologies, I used the incorrect name myself in my review of the original version of the manuscript)

Response: Has been corrected (main text: **line 76**).

In the abstract lines 27-28 it is stated that: "we accurately constructed a panel of 1,874 haplotypes haploid genomes with 41,964,356 genetic variants. However, in the results (lines 117-120) a larger number of genetic variants is mentioned: "Finally, 937 individuals ($N \geq 1$, No.breeds = 41; $N \geq 3$, No.breeds = 33) and 45,191,157 variants (41,326,535 SNPs; 3,864,622 Indels) were used for further analyses. Then in lines 181-183 it again is mentioned that: "The 937 high-quality individuals and 41,964,356 autosomal variants (38,483,119 SNPs; 3,481,237 Indels) were used for constructing a haplotype reference panel."

Response: We thank Reviewer #3 for this point. We constructed the haplotype reference panel using variants from 18 autosomes which harbor 41,964,356 genetic variants. But these 45,191,157 variants mentioned in the results include genetic variants from 18 autosomes and the X chromosome. Thus, the numbers of genetic variants in the abstract and the results differ. The subsequent analyses were not involved in variants from the X chromosome. To keep contextual statements consistent, we corrected the number "45,191,157 variants" to "41,964,356 autosomal variants" in the section Result (main text: **lines 115-116**).

Line 329: "suggests that Hap2 may originate from an extinct species (*Sus scrofa*)". "*(Sus scrofa)* should be deleted as this is not the suggested extinct species, but the recipient species.

Furthermore the alternative suggestion (line 330-331) "or occur at a too low frequency in other species to be detected by the small sample sizes used" seems highly unlikely.

Response: We thank Reviewer #3 for these helpful comments. The "*(Sus scrofa)*" has been deleted (main text: **lines 271**).

In the previous response for Reviewer #1 (please see pages 4-5 of the first response letter, specific comment 3), we tried to estimate the probability that seven samples from other species do not detect Hap2. But we failed to calculate this probability because the frequency of Hap2 in other species is unavailable. To investigate this problem further, we estimated the probability using the assumed frequency of Hap2 in other species as

follows:

- 1) Assume that the averaged frequency of Hap2 in other species is f_{Hap2} ranged from 0 to 1, with a gradient of 0.01.
- 2) Because these species are diploid, the probability that seven samples do not detect Hap2 is $P = (1 - f_{\text{Hap2}})^{14}$.
- 3) With the varies of f_{Hap2} , the probability was showed as **Fig. R1**.

If Hap2 occurs with a frequency greater than 0.193 in other species, that seven samples do not detect Hap2 is a small probability event ($P < 0.05$). Thus, before the frequency of Hap2 in other species is obtained, we could not deny the possibility that Hap2 occurs at a too low frequency to be detected by the small sample sizes used.

Fig. R1 Estimating the probability that seven samples do not detect Hap2 using the assumed frequency of Hap2 in other species. The y-axis denotes the probability that seven samples from other species do not detect Hap2. The x-axis denotes the assumed frequency of Hap2 in other species.

Line 580-581: “again in consist with the characteristics of balancing selection”. This sentence is not correct change to either “again consistent with” or “again in agreement with”.

Response: Has been corrected (main text: **line 460**).

Reference

1. Xiang, T., Christensen, O. F., Vitezica, Z. G. & Legarra, A. Genomic evaluation by including dominance effects and inbreeding depression for purebred and crossbred performance with an application in pigs. *Genet. Sel. Evol.* **48**, 1-14 (2016).
2. Reynolds, E. G., *et al.* Non-additive association analysis using proxy phenotypes identifies novel cattle syndromes. *Nat. Genet.* **53**, 949-954 (2021).

REVIEWERS' COMMENTS

Reviewer #1 (Remarks to the Author):

The authors have addressed the few remaining issues in this revised version. I have only a couple of comments on the parts that have been revised in this version.

1. You need to revise the following sentence on Line 272-274:

"The Hap1 were clustered with Sumatra wild boar, Visayan warty pig, and Javan warty pig, indicating their older age than Hap2 and the emergence of Hap1 before the divergence of the *Sus* species."

Hap1 is not older than Hap2, they are equally old. Figure 2F indicates that Hap1 and Hap2 split a long time ago and thus must have the same age. You may also change the text as follows "Hap1 haplotypes clustered with..." to avoid the problem whether Hap1 should be considered singularis or pluralis

2. Table 1: there are a few things that need to be fixed in this table:

- Line 326: Latin names should be in italics

- Line 313 and 327: I assume OR is not an Odd ratio but an Odds ratio

- I think it is better to present the incidence of ETEC as a percentage, 0.67 and 0.35, in particular since you report the additive and dominance effects as percentages. Also, I assume you compare additive effects as Hap1 - Hap2, then the additive effect for ETEC should be a positive value, higher incidence associated with Hap1?

- The significance values for the Additive and Dominance effects look fishy. For instance, for carcass length the P value is reported as $10e-7$ but the Additive effect is reported as "only" $<P=0.01$ and the dominance effect as non-significant. If there is no dominance effect then the additive effect should explain the great majority of the effect on the phenotype. The authors should check that the significances of the additive (and perhaps dominance) effects have been calculated correctly.

- Line 355: change to ", a new variable was constructed,"

- Line 377: change to P value to be consistent with the Table head and line 328

Response to Reviewer #1

May 13, 2023

We carefully checked the comments and revised the paper by point to point. All revisions were highlighted in red in the manuscript with track. The point-by-point responses to the concerns are listed as follows.

Responses to Reviewer #1

The authors have addressed the few remaining issues in this revised version. I have only a couple of comments on the parts that have been revised in this version.

Response: We greatly thank Reviewer #1 for his/her precious time in reviewing our work again.

1. You need to revise the following sentence on Line 272-274:

"The Hap1 were clustered with Sumatra wild boar, Visayan warty pig, and Javan warty pig, indicating their older age than Hap2 and the emergence of Hap1 before the divergence of the *Sus* species." Hap1 is not older than Hap2, they are equally old. Figure 2F indicates that Hap1 and Hap2 split a long time ago and thus must have the same age. You may also change the text as follows "Hap1 haplotypes clustered with..." to avoid the problem whether Hap1 should be considered singularis or pluralis.

Response: We thank Reviewer #1 for pointing this out. We have revised the sentence to "Hap1 haplotypes clustered with Sumatra wild boar, Visayan warty pig, and Javan warty pig, indicating the emergence of Hap1 before the divergence of the *Sus* species." (main text: **lines 245-247**).

2. Table 1: there are a few things that need to be fixed in this table:

- Line 326: Latin names should be in italics

Response: We have corrected "Escherichia coli" to "*Escherichia coli*" (main text: **lines 545 and 723**).

- Line 313 and 327: I assume OR is not an Odd ratio but an Odds ratio

Response: Has been corrected (main text: **lines 262 and 724**).

- I think it is better to present the incidence of ETEC as a percentage, 0.67 and 0.35, in particular since you report the additive and dominance effects as percentages. Also, I assume you compare additive effects as Hap1 - Hap2, then the additive effect for ETEC should be a positive value, higher incidence associated with Hap1?

Response: We thank Reviewer #1 for this comment. We have corrected the presentation of ETEC occurrence from "476(+)/235(-)" to an incidence of "0.67" and from "286(+)/529(-)" to an incidence of "0.35" in **Table 1**.

For the additive effect for ETEC, we checked the calculation step again. We selected a proxy SNP 13_135417839, its allele A (G) in complete linkage disequilibrium with Hap1 (Hap2), to estimate the additive effect of the haplotype of *MUC13* with model $Y = B + A * X1 + D * X2$. *B* denotes intercept. Coefficient *A* represents the estimated additive effect. *X1* represents the genotypes (A/A, A/G, G/G) of SNP 13_135417839. Genotypes A/A, A/G, and G/G were recoded as 2, 1, and 0, respectively. *Y* denotes the

phenotypic value 0 (susceptible to ETEC F4ac) or 1 (resistance to ETEC F4ac). In this way, we obtained a negative value for the additive effect of ETEC. If we represent the phenotypic value 0 (resistance to ETEC F4ac) or 1 (susceptible to ETEC F4ac), the additive effect for ETEC is a positive value. Thus, the sign of the additive effect for ETEC depends on how the phenotype is coded. But it does not influence the fact that a high incidence of ETEC F4ac is associated with Hap1.

To present it clearly, we coded resistance as 0 and susceptibility as 1. The sign of the additive effect and dominance effect for ETEC was changed (**Table 1**). The corresponding footnotes have been corrected as the following: N, sample size; ETEC F4ac adhesion, susceptible to Enterotoxigenic *Escherichia coli* F4ac; OR, Odds ratio; Neck Bone Length, total length of cervical vertebra; Hap1 or Hap2, haplotypes of *MUC13*. The significance (*P* Value) of ETEC F4ac phenotypic difference between Hap1 and Hap2 was calculated by Pearson's Chi Squared test. The significances (*P* Value) of remaining phenotypic differences between Hap1 and Hap2 was calculated by Student's *t* test (two-sided). Reported log transformed *p*-values are nominal (i.e. not corrected for multiple testing). Proxy SNP 13_135417839, its allele A (G) in complete linkage disequilibrium with the haplotype Hap1 (Hap2) of *MUC13*, was used to estimate the additive and dominance effects with model $Y = B + A * X1 + D * X2$. *Y* denotes the phenotypic value as 0 (resistance to ETEC F4ac) or 1 (susceptible to ETEC F4ac). *B* denotes intercept. *X1* represents the genotypes (A/A, A/G, G/G) of SNP 13_135417839. A/A, A/G, and G/G were recoded as 2, 1, and 0. Coefficient *A* represents the estimated additive effect. To account for dominance, a new variable was constructed, *X2*, with values of 0, 1, and 0 for, respectively, genotypes A/A, A/G, and G/G. Coefficient *D* represents the estimated dominance effect. *** represents that the effect is highly significant. The *P* values for the additive effects of ETEC F4ac, carcass straight length, carcass diagonal length, small intestine length, fourth cervical vertebra length, fifth cervical vertebra length, sixth cervical vertebra length, seventh cervical vertebra length, and neck bone length are 5.9×10^{-40} , 1.9×10^{-07} , 3.7×10^{-06} , 3.6×10^{-05} , 1.2×10^{-06} , 5.9×10^{-07} , 4.2×10^{-05} , 5.6×10^{-09} , and 1.2×10^{-04} , respectively. The *P* value for the dominance effect of ETEC F4ac is 5.2×10^{-05} .

Table 1 The phenotypic difference between haplotypes of *MUC13*.

Trait	Hap1		Hap2		P Value	Additive effect	Dominance effect
	N	Incidence Or Mean \pm SD	N	Incidence Or Mean \pm SD			
ETEC F4ac adhesion	711	0.67	815	0.35	4.1×10^{-35} (OR = 3.75)	0.328***	0.130**
Carcass straight length (cm)	854	97.04 \pm 6.87	1000	95.31 \pm 7.52	2.3×10^{-07}	1.823***	0.255
Carcass diagonal length (cm)	854	80.71 \pm 5.89	1000	79.39 \pm 6.41	3.9×10^{-06}	1.384***	0.249
Small intestine length (m)	856	15.88 \pm 1.93	1002	15.46 \pm 2.33	3.2×10^{-05}	0.429***	0.075
Fourth cervical vertebra length (cm)	854	2.03 \pm 0.20	1002	1.98 \pm 0.20	2.2×10^{-06}	0.048***	0.001
Fifth cervical vertebra length (cm)	853	1.99 \pm 0.19	997	1.95 \pm 0.18	9.2×10^{-07}	0.045***	0.007
Sixth cervical vertebra length (cm)	851	2.07 \pm 0.20	995	2.03 \pm 0.20	4.9×10^{-05}	0.040***	0.006
Seventh cervical vertebra length (cm)	851	2.29 \pm 0.21	999	2.23 \pm 0.21	1.0×10^{-08}	0.059***	0.004
Neck Bone Length (cm)	853	17.32 \pm 1.41	1003	17.06 \pm 1.49	9.1×10^{-05}	0.272***	0.109

N, sample size; ETEC F4ac adhesion, susceptible to Enterotoxigenic *Escherichia coli* F4ac; OR, Odds ratio; Neck Bone Length, total length of cervical vertebra; Hap1 or Hap2, haplotypes of *MUC13*. The significance (*P* Value) of ETEC F4ac phenotypic difference between Hap1 and Hap2 was calculated by Pearson's Chi Squared test. The significances (*P* Value) of remaining phenotypic differences between Hap1 and Hap2 was calculated by Student's t test (two-sided). Reported *P* Values are nominal (i.e. not corrected for multiple testing). Proxy SNP 13_135417839, its allele A (G) in complete linkage disequilibrium with the haplotype Hap1 (Hap2) of *MUC13*, was used to estimate the additive and dominance effects with model $Y = B + A * X1 + D * X2$. *Y* denotes the phenotypic value as 0 (resistance to ETEC F4ac) or 1 (susceptible to ETEC F4ac). *B* denotes intercept. *X1* represents the genotypes (A/A, A/G, G/G) of SNP 13_135417839. A/A, A/G, and G/G were recoded as 2, 1, and 0. Coefficient *A* represents the estimated additive effect. To account for dominance, a new variable was constructed, *X2*, with values of 0, 1, and 0 for, respectively, genotypes A/A, A/G, and G/G. Coefficient *D* represents the estimated dominance effect. *** represents that the *P* Value is highly significant. The *P* Values for the additive effects of ETEC F4ac, carcass straight length, carcass diagonal length, small intestine length, fourth cervical vertebra length, fifth cervical vertebra length, sixth cervical vertebra length, seventh cervical vertebra length, and neck bone length are 5.9×10^{-40} , 1.9×10^{-07} , 3.7×10^{-06} , 3.6×10^{-05} , 1.2×10^{-06} , 5.9×10^{-07} , 4.2×10^{-05} , 5.6×10^{-09} , and 1.2×10^{-04} , respectively. The *P* Value for the dominance effect of ETEC F4ac is 5.2×10^{-05} .

- The significance values for the Additive and Dominance effects look fishy. For

instance, for carcass length the P value is reported as 10e-7 but the Additive effect is reported as “only” <P=0.01 and the dominance effect as non-significant. If there is no dominance effect then the additive effect should explain the great majority of the effect on the phenotype. The authors should check that the significances of the additive (and perhaps dominance) effects have been calculated correctly.

Response: We thank Reviewer #1 for this comment. The *P* values for the additive effects of ETEC F4ac, carcass straight length, carcass diagonal length, small intestine length, fourth cervical vertebra length, fifth cervical vertebra length, sixth cervical vertebra length, seventh cervical vertebra length, and neck bone length were 5.9×10^{-40} , 1.9×10^{-07} , 3.7×10^{-06} , 3.6×10^{-05} , 1.2×10^{-06} , 5.9×10^{-07} , 4.2×10^{-05} , 5.6×10^{-09} , and 1.2×10^{-04} , respectively. The *P* value for the dominance effects of ETEC F4ac was 5.2×10^{-05} . Statistically, a *P* value less than 0.01 is considered highly significant. In **Table 1**, we used two asterisks **, representing *P* value < 0.01, to determine whether the additive and dominance effects were significant. To eliminate as much misunderstanding as possible, we added the concrete *P* values to footnotes of the **Table 1**. We described the significance of the additive and dominance effects with three asterisks *** and corrected the sentence "*** represents that p-value is less than 0.01." to "**** represents that the *P* Value is highly significant. The *P* Values for the additive effects of ETEC F4ac, carcass straight length, carcass diagonal length, small intestine length, fourth cervical vertebra length, fifth cervical vertebra length, sixth cervical vertebra length, seventh cervical vertebra length, and neck bone length are 5.9×10^{-40} , 1.9×10^{-07} , 3.7×10^{-06} , 3.6×10^{-05} , 1.2×10^{-06} , 5.9×10^{-07} , 4.2×10^{-05} , 5.6×10^{-09} , and 1.2×10^{-04} , respectively. The *P* Value for the dominance effect of ETEC F4ac is 5.2×10^{-05} ." (main text: **lines 735-741**).

- Line 355: change to “, a new variable was constructed,”

Response: Has been corrected (main text: **line 734**).

- Line 377: change to P value to be consistent with the Table head and line 328.

Response: If we understand correctly, Reviewer #1 means that the "p-value" at line 337 was not consistent with the "*P* Value" at the Table head and line 328. It now has been corrected to "**** represents that the *P* Value is highly significant." (main text: **line 736**).